# Interplay of structural chirality, electron spin and topological orbital in chiral molecular spin valves

Yuwaraj Adhikari[1,4], Tianhan Liu [1,4], Hailong Wang [2], Zhenqi Hua[1], Haoyang Liu[1], Eric Lochner[1], Pedro Schlottmann[1], Binghai Yan [3] ✉, Jianhua Zhao[2] ✉ & Peng Xiong [1] ✉

Chirality has been a property of central importance in physics, chemistry and biology for more than a century. Recently, electrons were found to become spin polarized after transmitting through chiral molecules, crystals, and their hybrids. This phenomenon, called chirality-induced spin selectivity (CISS), presents broad application potentials and far-reaching fundamental implications involving intricate interplays among structural chirality, topological states, and electronic spin and orbitals. However, the microscopic picture of how chiral geometry influences electronic spin remains elusive, given the negligible spin-orbit coupling (SOC) in organic molecules. In this work, we address this issue via a direct comparison of magnetoconductance (MC) measurements on magnetic semiconductor-based chiral molecular spin valves with normal metal electrodes of contrasting SOC strengths. The experiment reveals that a heavy-metal electrode provides SOC to convert the orbital polarization induced by the chiral molecular structure to *spin* polarization. Our results illustrate the essential role of SOC in the metal electrode for the CISS spin valve effect. A tunneling model with a magnetochiral modulation of the potential barrier is shown to quantitatively account for the unusual transport behavior.

Helical textures and monopole-like chirality in electronic structures of topological materials have given rise to a plethora of emergent phenomena characterized by unusual interplays between electronic charge, spin, and orbital[1–4]. More recently, a parallel phenomenon in real space, in which structural chirality induces electron spin polarization in the direction of their momentum, has received increasing attention[5–7]. The effect, termed chirality-induced spin selectivity (CISS), was first evidenced by Mott polarimetry of photoelectrons from a nonmagnetic (NM) Au electrode through a self-assembled monolayer (SAM) of short synthetic molecules of dsDNA[8]. Since then,

CISS has been observed in a variety of chiral molecular systems including macro[8–13] and small[14,15] molecules, supramolecular polymers[16], metal-organic frameworks[17], and hybrid organic-inorganic perovskites[18,19] and artificial superlattices[20,21], via a host of electrical, optical, and electrochemical probes[22–25]. More broadly, CISS is shown to effect enantio-selective chemical reactions[26] and facilitate enantiomer separation[27], and the adsorption of chiral molecules on the surface of a conventional superconductor was reported to induce unconventional superconductivity[28,29]. All these experiments suggest a highly consequential interaction between

[1]Department of Physics, Florida State University, Tallahassee, FL 32306, USA. [2]State Key Laboratory of Superlattices and Microstructures, Institute of Semiconductors, Chinese Academy of Sciences, 100083 Beijing, China. [3]Department of Condensed Matter Physics, Weizmann Institute of Science, Rehovot, Israel. [4]These authors contributed equally: Yuwaraj Adhikari, Tianhan Liu. ✉e-mail: binghai.yan@weizmann.ac.il; jhzhao@semi.ac.cn; pxiong@f-su.edu

molecular structural chirality and electronic spin, which carries profound and broad implications.

Despite increasing preponderance of experimental results and a great deal of theoretical efforts, the microscopic origin and physical mechanisms behind CISS remain open questions[30,31]. A central unsettled issue is the role of spin-orbit coupling (SOC) in the chiral media. SOC is a necessary element in the emergence of spin polarization in *NM* materials in general. Specifically, it is an essential ingredient in most theoretical models of CISS, whereas the SOC in the molecular materials is too weak to account for the experimentally significant spin selectivity at room temperature[32]. In order to overcome this difficulty, a number of theoretical approaches were proposed, based primarily on spin dependent scattering and tight-binding models[33–41]. The approaches have targeted at amplification of SOC, either its *value*, to account for the experimentally observed CISS spin polarization by introducing various factors such as density of scattering centers[36], dephasing[37,42], and environmental nonunitary effects[41], or its *effect*, through electron-electron correlation[43], exchange interactions[44], vibrational and polaronic effects[45,46], frictional dissipation[47], and Berry force[48].

An alternative approach rids of reliance on SOC in chiral molecules altogether[39,49,50]. Gersten et al.[39] introduced the concept of "induced spin filtering": A selectivity in the transmission of the electron orbital angular momentum can induce spin selectivity in the transmission process, provided that there is strong SOC in the substrate supporting the chiral SAM. This proposal, however, was questioned because CISS was observed in photoemission experiments in which the substrates have negligible SOC[11,15]. Liu et al.[49] noted an important difference between the manifestations of the CISS effect in photoemission setups[8,15] and transport in molecular junctions:[51,52] The former measures the "global orbital angular momentum" that includes both the orbital and spin angular momenta, whereas the latter probes spin polarization exclusively. Physically, the model suggests that chiral molecules act as an orbital filter rather than a spin filter, and the SOC in the metal electrode converts the orbital polarization into spin polarization, thus producing CISS without the need for any SOC in the molecules (see the illustration in Fig. 1a). The orbital polarization effect, which is caused by the orbital-momentum locking - an intrinsic topological property of electronic states in a chiral material[49,53], has much

broader relevance beyond CISS; in particular, it presents a new pathway for spin manipulation through atomic structure engineering. So far, however, definitive experimental evidence of the effect is lacking.

One of the most widely used device platforms to detect and utilize spin polarization are spin valves. In a CISS spin valve, the spin polarization of a charge current from the NM electrode through the chiral SAM is analyzed by the magnetic electrode, and the junction conductance is expected to depend on the magnetization direction of the magnetic electrode, resulting in a MC. For CISS studies, a scanning probe rendition of the spin valve, magnetic conductive atomic force microscopy (mc-AFM), has been frequently used[51]. Although its implementation is relatively straightforward, mc-AFM relies on large number of averaging to mitigate the fluctuation and instability. In contrast, thin film-based molecular junctions, in which a chiral SAM is sandwiched between two conducting electrodes, are more conducive to stable and reproducible current-voltage (I-V) and MC measurements. Such devices, however, present significant technical challenges of their own: Pinholes are almost always present in SAMs at device scales (~ μm), and in the cases of two metal electrodes, any direct contact will short out the device.

We recently demonstrated that these complications can be effectively mitigated by using a ferromagnetic semiconductor, (Ga,Mn)As, as the magnetic spin analyzer[52]. The use of the magnetic semiconductor was found to alleviate the shorting problem due to the presence of a Schottky barrier at direct contact with the Au electrode. Moreover, the (Ga,Mn)As was grown by molecular beam epitaxy (MBE) on an (In,Ga)As buffer layer, and the resulting strain from the lattice mismatch produces perpendicular magnetic anisotropy (PMA)[54], namely an out-of-plane magnetization that is collinear with the spin polarization from CISS. These two unique device characteristics enabled observation of spin-valve MC distinctly associated with CISS, and the inherent stability of the platform facilitated a first rigorous determination of the bias current dependence of the MC from CISS[52].

Leveraging this proven device platform, we fabricated and characterized a deliberately chosen set of (Ga,Mn)As/SAM/NM heterojunctions. The experiments yielded quantitative differentiation of the magnitude and bias-dependence of the spin valve conductance in junctions with NM electrodes of contrastingly different SOC strengths

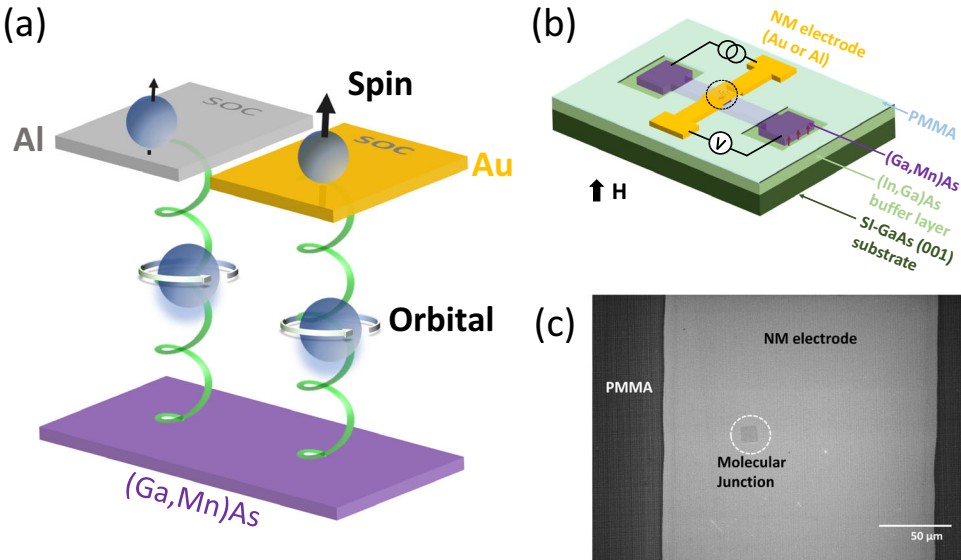

**Fig. 1 | The chiral molecular spin valve and orbital to spin conversion mechanism. a** Schematic depiction of the mechanism for orbital to spin polarization conversion in a (Ga,Mn)As/chiral molecule/NM spin valve. Chiral molecules induce orbital polarization in passing electrons and subsequently, the electrode

SOC converts the orbital to spin (represented by the black vertical arrows on the top electrodes)[49]. **b** Schematic diagram of the device structure along with the junction measurement setup. **c** A top-view scanning electron microscopy image of the junction region (black circle in (**b**)) in a molecular spin-valve device.

(Au versus Al) and SAMs of chiral and achiral molecules. The results revealed a definitive correlation between the magnitude of the CISS spin valve conductance and the SOC strength in the NM electrode: The molecular junctions with Au electrodes exhibit significant MC whose magnitudes depend distinctly on the chirality or length of the molecules; in contrast, in otherwise identical devices with Al electrodes, regardless of the molecule involved the MC are essentially indistinguishable from those of the control samples without any molecules. A model based on magnetochiral modulation of the tunneling barrier potential[55] from orbital polarization is shown to provide quantitative account for both the magnitude and bias dependence of the MC of the two types of junctions. The work unambiguously evidenced the essential role of the contact SOC in generating observable CISS effect in chiral molecular spin valves.

## Results and discussion

### Chiral molecular spin valve and orbital to spin conversion

We detect the MC in chiral molecular spin valve devices with a (Ga,Mn)As magnetic electrode and Au or Al normal metal electrode. Figure 1a shows a schematic diagram depicting the molecular junction structure and the physical mechanism for the chirality-induced orbital polarization and subsequent orbital to spin polarization conversion due to the SOC in the NM electrode. Figure 1b is schematics of the device heterostructure and setups for the quasi-four-terminal I-V and conductance measurements. Figure 1c is an SEM image of a junction; the junctions were squares of sizes ranging from $5 \times 5\,\mu m^2$ to $15 \times 15\,\mu m^2$.

Molecular assembly, the formation of the SAM on (Ga,Mn)As, is a critical step in the device fabrication process. For this work, alpha-helix

L-polyalanine (AHPA-L) and L-cysteine served to compare chiral molecules of different molecular lengths, whereas 16-mercaptohexadecanoic acid (MHA) and 1-octadecanethiol (ODT) were used as achiral molecules of similar length but with different terminal groups. The total molecular length of AHPA-L is 5.4 nm, whereas L-cysteine is around 4 Å, and those of MHA and ODT are 2.4 nm and 2.7 nm, respectively. All four molecules contain a thiol end group, which facilitates formation of high-quality SAMs on the (Ga,Mn) As[52,56,57]. The experimental details are described in "Methods" section.

As described previously[52], despite the probable presence of direct contacts between the NM and (Ga,Mn)As (parallel conduction) through defects in the SAM, the spin valve conductance due to CISS in these junctions can be identified from the difference in junction conductance, ΔG, under opposite saturation magnetization for the (Ga,Mn)As. Figure 2a, b shows representative sets of MC measurements with varying perpendicular magnetic field for (Ga,Mn)As/AHPA-L/NM junctions with NM electrode of Au and Al, respectively, measured at various constant bias currents at T = 4.8 K. Each MC curve shows two distinct conducting states, coinciding with the well-defined square magnetic hysteresis of the (Ga,Mn)As due to its PMA, hence a ΔG can be precisely determined. As expected, the square hystereses of the MC resemble those of the anomalous Hall effect (AHE) of the (Ga,Mn)As which is directly proportional to M. Any significant contribution of AHE of the (Ga,Mn)As in the junction MC was ruled out in our previous work[52]. Here we present an updated list of justifications in Supplementary Information (Sec. 5).

The square hysteresis thus facilitates a straightforward and reliable determination of detailed bias current dependence of ΔG, from

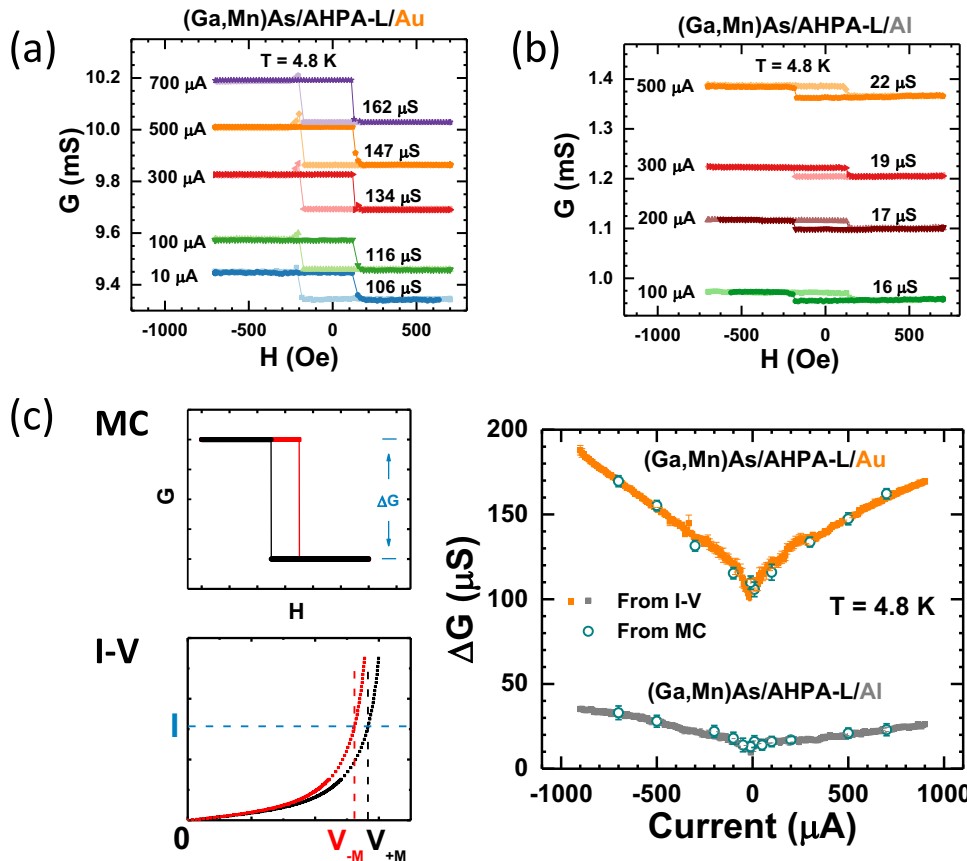

**Fig. 2 | CISS spin valve conductance: effect of the NM electrode.** Representative MC curves measured at different bias currents for (Ga,Mn)As/AHPA-L/NM junctions, with NM of (**a**) Au, (**b**) Al. **c** The bias dependence of ΔG for the junctions with Au and Al contact. The solid squares are measured from I-V curves, whereas empty circles are measured from MC measurements at different bias currents. The left panels illustrate how the values of ΔG are extracted from MC and I-V measurements.

I-V curves measured under the opposite magnetization states of the (Ga,Mn)As, as shown in Fig. 2c. Specifically, the $\Delta G$ from the MC measurements can be obtained from and corroborated by the I-V's as[52]

$$\Delta G = I\left(\frac{1}{V_{-M}} - \frac{1}{V_{+M}}\right), \qquad (1)$$

where $V_{+M}$ and $V_{-M}$ indicate the corresponding bias voltage upon switching the magnetization in the (Ga,Mn)As from $+M$ to $-M$ at the same current $I$. Figure 2c shows $\Delta G$ as functions of bias current for the junction; as expected, the two types of measurements produced consistent results. The I-V data are presented in Supplementary Fig. 1. The same measurement and analysis procedures were applied to all devices with different molecules and NM electrodes in this study to obtain $\Delta G$ and their bias current dependences.

We note that the total junction conductance (G) for the Au junction is also much greater than that of the Al junction. Nevertheless, measurements on the large number of (Ga,Mn)As/AHPA-L/Au devices clearly demonstrated that although the I-V and total conductance may vary greatly in the molecular junctions of similar structures depending on the degree of parallel conduction (quality of the SAM assembly), both the magnitude and bias current dependence of the CISS spin valve conductance ($\Delta G$) were found to be consistently similar. A detailed discussion and comparison with a Au junction shown in our previous work[52] are presented in Supplementary Information (Sec. 2). We conclude that the total G is spurious (depending the molecular coverage of the SAM) and has no bearing on CISS spin-valve conductance; it is the $\Delta G$ that accurately reflects the CISS effect. The fact that different Au junctions show large variations of the I-V and total G, but exhibit similar bias-dependent $\Delta G$, lends further credence to our model and the associated analyses and conclusion.

### Effect of the NM electrode
Figure 2c shows CISS spin valve conductance for two (Ga,Mn)As/AHPA-L/NM junctions with Au and Al as the NM electrode. The experiment constitutes a direct comparison of the magnitude and bias dependence of $\Delta G$ for two NM electrodes of contrasting SOC strengths. The most notably result here is the pronounced differences between the junctions with Au and Al electrodes. Figure 3 shows the results from a comparative experiment with AHPA-L replaced by the much shorter chiral molecule of L-cysteine. As expected, with the same Au electrode, $\Delta G$ for the L-cysteine junctions are significantly smaller than those of the AHPA-L counterparts[51,58]. Remarkably, the large difference between $\Delta G$ is also present for the L-cysteine junctions with Au and Al

electrodes. We emphasize that for each combination of chiral SAM and NM electrode, the entire set of measurements was repeated in multiple samples (2 to 4), each with 4 junctions, and the results were consistent. The experiments, therefore, strongly indicate that the observed significant impact of the NM electrodes on the CISS spin valve effect originates from inherent differences of Au and Al, and is independent of the specific chiral molecule used.

We have also fabricated and measured large numbers of devices with Cu and Ag electrodes, two NM materials of intermediate SOC strengths between Au and Al. However, despite the repeated attempts, for either material, we were unable to obtain results with the degree of consistency achieved in Au and Al devices. Most Ag and Cu junctions yielded very small $\Delta G$ without the regular bias dependences. We surmise that this was due to poor interfaces or even damages to the molecular SAM by Cu and Ag. As shown in Supplementary Fig. 3, one Cu junction yielded $\Delta G$ that fits well between those of the Au and Al junctions, however, it does not exhibit the bias-current dependence consistently seen in Au and Al junctions. For these reasons, we are unable to make a definitive statement regarding the CISS spin valve conductance and SOC strengths in Cu and Ag at this point.

### Chiral versus achiral molecules
We further examined the spin valve effect in the molecular junctions with achiral molecules. Figure 4 shows the variation of the spin valve signal from chiral to achiral molecules in the molecular junctions with the same NM electrode of Au (Fig. 4a) and Al (Fig. 4b). In the Au

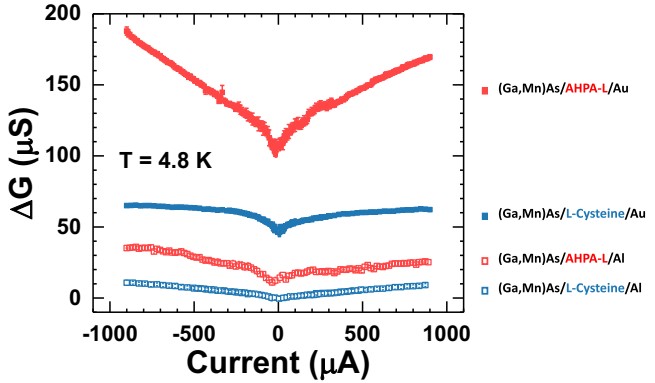

**Fig. 3 | CISS spin valve conductance: chiral molecules of different lengths.** Bias current dependences of $\Delta G$ for four junctions of (Ga,Mn)As/AHPA-L (L-cysteine)/Au (Al).

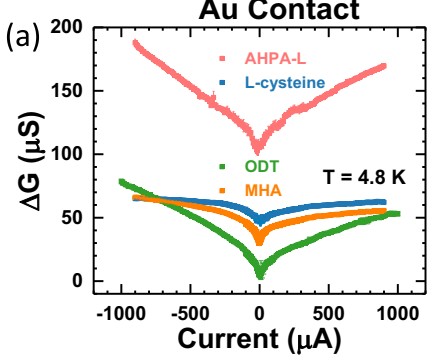

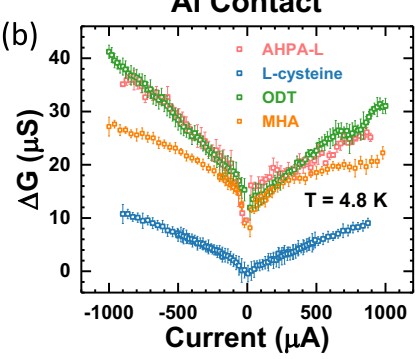

**Fig. 4 | CISS spin valve conductance: chiral versus achiral molecules.** Bias current dependences of $\Delta G$ for the molecular junctions of different chiral (AHPA-L and L-cysteine) and achiral (MHA and ODT) molecules with (**a**) Au, and (**b**) Al contact.

The pink and blue squares are the chiral molecular junctions (AHPA-L and L-cysteine), whereas green and orange squares are from the molecular junctions with achiral molecules (MHA and ODT).

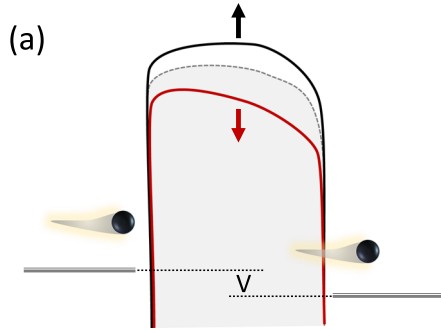

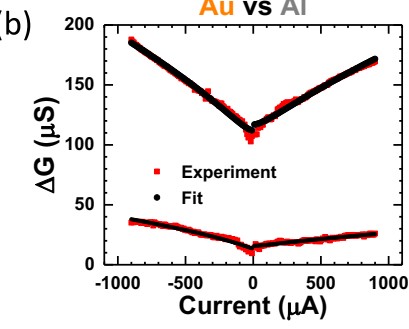

**Fig. 5 | Barrier tunneling analysis of the CISS spin valve conductance.** The bias dependent $\Delta G$ for the chiral molecular junctions can be fitted to a model of magnetochiral modulation of the tunneling barrier. **a** Schematic depiction of the tunneling model and barrier modulation mechanism. Red and black curves illustrate the modified tunneling barriers by $\downarrow$ and $\uparrow$ magnetizations, respectively, in the (Ga,Mn)As electrode. The original barrier is represented by the gray dashed curve. The bias voltage is indicated by V. **b** Fitting results of the spin valve conductance for two (Ga,Mn)As/AHPA-L/NM junctions with Au and Al contact.

junctions, there is a clear difference in the magnitude of the current-dependent $\Delta G$ between the chiral molecule (AHPA-L) and achiral molecules (MHA and ODT). It is important to note that despite the much-diminished magnitude, $\Delta G$ in the achiral molecular junctions are not trivial. Their magnitudes are clearly above the background in the control junction; in fact, they are comparable to $\Delta G$ in the short chiral molecule (L-cysteine) junctions. Moreover, $\Delta G$ in the MHA and ODT junctions exhibit distinct bias dependences resembling those in the chiral molecular junctions. These observations are consistent with a prediction from the orbital polarization model that nontrivial spin transport can materialize even in non-helical and even achiral systems, because in the presence of time-reversal symmetry, the emergence of orbital texture requires only inversion symmetry breaking[49], not necessarily chirality. Both MHA and ODT molecules have a thiol end and different terminal groups, thus are polar and possess inversion asymmetry. Here, the Al junctions again provide an illuminating comparison (Fig. 4b): There are no discernible differences in the $\Delta G$ for the AHPA-L and achiral molecular junctions. In fact, as shown in Supplementary Fig. 5, regardless of the molecule involved, the MC of all the Al junctions are comparable to that of the control junction without any molecules. In previous[52] and current work, we have conducted extensive measurements on large number of control samples; a representative set of results are shown in Supplementary Information (Sec. 4). It is clear that the MC in the control samples were close to the noise level and lacking any well-defined bias dependence, thus constituting a viable reference for meaningful comparison of the MC in various molecular junctions.

## Tunneling model and magnetochiral modulation of potential barrier

Summarizing the key experimental observations, several robust features have been conclusively identified from our experiments: An optimal two-terminal chiral molecular spin valve, as exemplified by the (Ga,Mn)As/AHPA-L/Au junction, consistently exhibits significant MC, which increases linearly with the bias current at high biases but approaches a finite value at zero-bias. Contrary to expectation from the Onsager reciprocal relation[49,59,60], the MC is symmetric (i.e. the sign of $\Delta G$ remains the same), rather than anti-symmetric, upon reversal of the current direction. Moreover, the magnitude of the MC decreases precipitously when AHPA-L is replaced by a much shorter chiral molecule or achiral molecules. Most importantly, replacing the Au electrode by Al results in even greater reduction of the MC, to levels where the differences between junctions of the different molecules are no longer discernible. Taken together, these results have revealed valuable new insights and placed several important constraints on a viable mechanism of CISS.

The orbital polarization model[49] offers a natural account for the observed qualitative differences between the molecular junctions with Au and Al electrodes. More recently, incorporating orbital polarization, a tunneling model was proposed to describe the electronic transport in chiral molecular junctions[55]. The essential ideas are depicted in Fig. 5a: The molecular chirality and electrode magnetism modulates tunneling barrier through the insulating molecular junction, termed magnetochiral modulation, which originates from the magnetochiral anisotropy[55]. Physically, a current flow induces charge accumulation, which changes the tunneling barrier in a way that depends on the magnetization and molecular chirality. We demonstrate that this model[55] can provide a semi-quantitative self-consistent description of all the key observations in the following.

We incorporate the magnetochiral modulation into the Simmons model[61,62] of metal/insulator/metal tunneling junctions. The problem of an arbitrarily shaped potential barrier is modeled into that of a rectangular barrier, which results in an explicit expression for the $I(V)$. The Simmons model and its variants have been widely applied to the modeling of electron transport in molecular junctions[63,64]. Based on the Simmons expression and assuming a small magnetochiral modulation of the potential barrier, in the intermediate bias range, we approximate the magnetization-dependent differential conductance through a chiral molecular junction in the form of a simple exponential relation:

$$\frac{dI}{dV} = (\alpha_o + \alpha_M)e^{\beta V_M} \tag{2}$$

where $\alpha_o$ and $\beta$ are magnetization-independent coefficients reflecting the tunneling current and probability across the unmodified potential barrier, and $\alpha_M$ is the coefficient that quantifies the effect from the change of the potential barrier height upon switching of the magnetization of the (Ga,Mn)As, namely the CISS spin valve conductance $\Delta G$. The modulation of the potential barrier is small and considered a perturbation, $\alpha_M \ll \alpha_o$. $\Delta G$ as a function of the bias current (Eq. (1)) can then be evaluated as:

$$\Delta G(I) = \beta I \left[ \frac{1}{\ln\left(1 + \frac{\beta I}{\alpha_0 + \alpha_{-M}}\right)} - \frac{1}{\ln\left(1 + \frac{\beta I}{\alpha_0 + \alpha_{+M}}\right)} \right] \tag{3}$$

Figure 5b shows the best fits of the $\Delta G$ for the two (Ga,Mn)As/AHPA-L/NM junctions (NM = Au, Al) to Eq. (3). More details of the fitting procedure are presented in Sec. 6 of the Supplementary Information. In brief, the fitting is performed separately for positive and

**Table 1 | Fitting parameters for the tunneling model**

| $\alpha$ value ($\mu$S) | Positive current | | | Negative current | | |
|---|---|---|---|---|---|---|
| Junction | $\alpha_o$ | $\alpha_{-M}$ | $\alpha_{+M}$ | $\alpha_o$ | $\alpha_{-M}$ | $\alpha_{+M}$ |
| (Ga,Mn)As/ AHPA-L/Au | 1130 | 29.8 | –20.9 | 786 | 28.2 | –20.4 |
| (Ga,Mn)As/ AHPA-L/Al | 836 | 2.92 | –3.95 | 233 | 2.42 | –3.49 |

Values of $\alpha_o$, $\alpha_{-M}$ and $\alpha_{+M}$ from fittings of the CISS spin valve conductance data to Eq. (3). Here, $\beta$ is kept constant at 10 V$^{-1}$. The most notable result is that for Au and Al junctions, $\alpha_o$ are similar while $\alpha_{\mp M}$ differ by an order of magnitude.

negative currents. In a typical fitting process, an optimal value of $\beta$ is first identified. With $\beta$ fixed, best fit to the data to Eq. 3 is then performed with $\alpha_o$, $\alpha_{-M}$ and $\alpha_{+M}$ as the fitting parameters. Table 1 lists the resulting values for the parameters from the best fits.

Two notable features are evident in Table 1. First, $\alpha_o \gg \alpha_{\mp M}$, consistent with our assumption that the magnetochiral modulation of tunneling barrier is small, but the CISS-induced spin valve conductance is large due to the exponential dependence of the junction conductance on the barrier height. Furthermore, the magnitudes of $\alpha_{\mp M}$ in the junction with Au contact is an order of magnitude greater than those with Al electrode, while the values of $\alpha_o$ are similar in both junctions. The result constitutes quantitative support for the hypothesis that $\alpha_M$ is magnetization dependent and its strength depends on the SOC of the NM electrode. In addition, it is evident that the fittings provide excellent description of the observed bias current dependences of $\Delta G$, and naturally account for the fact that $\Delta G$ is essentially independent of current direction. The higher order asymmetry in the data is reflected in the different values of $\alpha_o$ for positive and negative current. We note that slight asymmetries in the tunneling conductance are commonly observed and expected in the Simmons model[62] for junctions with dissimilar metallic electrodes and/or asymmetric potential barriers. The fitting results for other junctions, including those of achiral molecules, are described in Sec. 7 of the Supplementary Information. The results provide a quantitative measure of the different effects of the normal metal electrode (Au versus Al) consistent with the qualitative trends of $\Delta G$ apparent in Figs. 2–4 and Supplementary Fig. 5: For the Au junctions $\alpha_M$ are large and show a consistent decrease from AHPA-L to the achiral molecules; in contrast, all Al junctions show much smaller $\alpha_M$ without any systematic difference depending on the chirality and length of the molecules.

The observation of CISS MC in the spin valves of the achiral molecules and their magnitudes are well accounted for in our model. The molecular polarity (asymmetry) factors directly into the magnetochiral modulation of the tunneling barrier[55]. Physically, the magnetochiral modulation of the tunneling barrier of the insulating molecules can be understood as an analog of the electrical magnetochiral anisotropy (eMChA) in conductors[65]. It is therefore important to note that *eMChA has been observed in both chiral and (nonchiral) polar conductors* (see Fig. 3 in ref. 66 for a comprehensive list). Quantitatively, the eMChA is stronger in a chiral crystal with well-defined structural chirality[67] than in a nonchiral polar solid. Therefore, finite spin-valve conductance in junctions of achiral (polar) molecules are expected in our model and their magnitudes should be smaller relative to that in chiral molecular junctions. Further theoretical work is needed to relate the quantitative differences with the molecular structures and physical parameters of Au and Al.

Finally, we comment on the presence of low-bias MC, most notably in the (Ga,Mn)As/AHPA-L/Au junctions. Experimentally, finite low-bias MC has been observed in a variety of two-terminal junctions of different chiral molecules and device structures[20,21,52,68,69]. On the other hand, theoretically, there is continuing debate as to whether and how this can be reconciled with the Onsager relation, which precludes

a linear-response MC in the two-terminal chiral spin valves[59,60,70,71]. This topic was a focal point of our previous work[52], here we describe a possible pathway out of the dilemma within the proposed model above. Because the model relies on bias current induced magnetochiral modulation of the tunneling barrier, any hysteresis in the charging/discharging process would yield a finite remnant ("linear-response") spin valve conductance even at zero bias. We note that bias-induced conductance switching, and I-V hysteresis have been reported in a variety of molecular junctions of different configurations and molecules[72–76]. Since the conductance is exponentially sensitive to the tunneling barrier, the induced spin valve conductance can be considerable even for a small remnant modulation (splitting) of the potential barrier. In one of the devices we reported on previously (Supporting Fig. 5, ref. 52), where particular care was taken to examine the effect of bias history, pronounced bias-induced changes in I-V and MC were observed. The results are shown in the current Supporting Information (Sec. 8). In this junction, for applied bias currents up to 800 $\mu$A, the junction showed negligible low-bias (linear-response) MC, whereas the application of a 1000 $\mu$A bias current induced substantial increases in MC over the full bias range; most notably, a significant zero-bias MC was now present (Supplementary Fig. 8). Concurrently, the I-V characteristics of the junction also saw significant changes due to the applied large bias (Supplementary Fig. 8b). An interesting contrasting example was reported in ref. 77, where CISS MC was measured in a long lateral two-terminal device. In such a device, the electron transport is likely to be diffusive instead of tunneling, and the zero-bias MC was vanishingly small.

In summary, utilizing a robust device platform of magnetic semiconductor-based molecular junctions proven effective for CISS studies, we have obtained direct experimental evidence that the SOC in the NM electrode is essential to the emergence of the CISS spin valve effect. With a Au electrode, the precipitous decrease of the spin valve conductance from AHPA-L junctions to those of shorter chiral molecule and achiral molecules is readily discerned. Replacing the Au electrode with Al results in pronounced drops of the spin valve conductance for all molecules, to the degree that the differences between the molecules and with the control junctions are no longer discernible. A model based on orbital polarization from inversion-symmetry breaking and the resulting magnetochiral modulation of the tunneling barrier potential is shown to not only consistently account for all key aspects of the experimental results, but also provide resolution to several long-standing open issues in the field, including finite low-bias magnetoresistance due to CISS in two-terminal molecular spin valves, the spin valve conductance being symmetric upon current reversal, and its magnitude often being much greater than what is expected from the spin polarization of the ferromagnet. Our work reveals a close relation between chirality and electronic properties, in which structural chirality information is transferred and transformed from molecular geometry to electronic orbital and eventually to the electronic spin via SOC. The results thus provide useful guidelines for detecting chirality-induced phenomena and designing CISS devices.

## Methods
### Materials
The AHPA-L in the experiments was obtained from RS Synthesis, LLC. L-cysteine, MHA, and ODT were acquired from Sigma-Aldrich, Inc. Molecules, except L-cysteine, were dissolved in pure ethanol at 1 mM concentration. L-cysteine was dissolved in deionized water at a concentration of 2.5 mM. The AHPA-L solution was kept at -18 °C for storage whereas L-cysteine, MHA and ODT solution were stored in ambient conditions.

The (Ga,Mn)As with perpendicular magnetic anisotropy was grown by low-temperature molecular-beam epitaxy (LT-MBE). 500 nm thick (In,Ga)As buffer layer was first grown at 450 °C on semi-insulating (001) GaAs substrates. (Ga,Mn)As films of 40 nm thickness with

perpendicular magnetic anisotropy were later grown at substrate temperature of 270 °C. The (Ga,Mn)As thin film has a Curie temperature of 80 K and coercive field of 174 Oe.

## Device fabrication process

The junction devices are fabricated in the following steps:

a. **Junctions Defined by Electron-Beam Lithography (EBL):** The sample was spin-coated with 2% PMMA at 4 krpm for 30 s. It was prebaked at 180 °C for 10 min on a hot plate. The EBL was performed to draw the junction patterns of size ranging from $5 \times 5\,\mu m^2$ to $15 \times 15\,\mu m^2$. The sample was developed in methyl isobutyl ketone (MIBK) diluted with isopropanol (1:3) for 40 s and then in pure isopropanol for 30 s at room temperature.

b. **Removal of oxide layer on (Ga,Mn)As and molecular assembly:** The sample was cleaned with $O_2$ plasma to remove any organic residue. It was set with medium power at 200 mTorr oxygen pressure for 1 min. The sample was then baked at 180 °C for 20 min on the hot plate to harden the PMMA. To remove the native oxide layer on the (Ga,Mn)As, the sample was etched with an ion mill for 1 min 30 sec. It was immersed in ethanol immediately after being taken out from the ion mill chamber before the molecular assembly. For the assembly of molecular monolayer on the (Ga,Mn)As, the sample was left in the molecular solution at room temperature for 24 h. After the assembly, the sample was rinsed with ethanol and dried with nitrogen gas. The molecular assembly process is similar for both chiral and achiral molecules.

c. **Top NM Electrodes Deposition:** A metallic shadow mask was used to draw the electrodes patterns which was aligned with the molecular junctions in a sample under optical microscope. Thermal evaporation was done to deposit NM electrodes. During the deposition of the top magnetic electrode, the substrate is cooled with liquid nitrogen and the temperature is maintained below −50 °C. 35 nm of Au (without Cr) and 50–70 nm of Al was deposited through a shadow mask.

## Electrical measurements

The sample was fixed on a socket with a copper base and wired with silver paint and Pt wire. All the measurements were performed in an Oxford ³He cryostat at the temperature range of 4.2–5.5 K. The measurement procedure is similar to previously reported[52]. Magnetic field perpendicular to the sample plane was applied up to 700 Oe. The (Ga,Mn)As was first magnetized at 700 Oe, and then the magnetic field was swept at a constant rate of 350 Oe/min for measurements. DC measurements were done with Keithley 2400 as the current source and HP 3458 as the voltmeter. AC measurements were performed with SR2124 dual-phase analog lock-in amplifiers.

## Data availability

The data supporting the findings of this study are available within the main text of this article and its Supplementary Information. Further information of this study is available from the corresponding authors upon reasonable request. The Source data are provided with this paper.

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

## Acknowledgements

We acknowledge helpful discussions with Hanwei Gao and Jiewen Xiao. The work at FSU is supported by NSF grant DMR-1905843 (P.X). The work at IOS is supported by the MOST grant 2021YFA1202200, the CAS Project for Young Scientists in Basic Research (YSBR-030) and the Strategic Priority Research Program of the Chinese Academy of Sciences under Grant No. XDB44000000 (H.W and J.Z). B.Y. acknowledges the financial support by the European Research Council (ERC Consolidator Grant "NonlinearTopo", No. 815869), the MINERVA Stiftung with the funds from the BMBF of Germany, and the Israel Science Foundation (ISF No. 2932/21).

## Author contributions

P.X., B.Y., T.L. and Y.A. conceived the project. Y.A. fabricated the devices and performed experiments, assisted by T.L., Z.H., H.L. and E.L. The MBE growth of materials was performed by H.W. and J.Z. P.X. and J.Z. supervised the project. P.X., B.Y., Y.A. and Z.H. performed theoretical modeling. Y.A., T.L., P.X. and B.Y. wrote the manuscript with input from all authors.

## Competing interests

The authors declare no competing interests.
