## [Peer Review File · Nature Communications]

Reviewers' Comments:

Reviewer #1:

Remarks to the Author:

The authors have performed spin valve magneto resistance measurements where they have assembled chiral and achiral molecules on magnetic (Ga, Mn)As. They observe a change in the MR modulation of the conductance when they compare Au and Al top electrodes. They interpret their results in terms of the role of the different strengths of the spin orbit interaction (SOC). They provide a theoretical model.

I cannot recommend publication. The most important reason is that the authors observe MR even for achiral molecules, and even without any molecules at all. This strongly suggest that the MR does not originate from the molecular junction itself (due to CISS related effects) , but comes from the magneto resistance (e.g. anomalous Hall effect, or related effects) of the (Ga,Mn) As layer. Despite the crucial role of this layer, (almost) nothing is mentioned about the characterization of the magnetic behavior this layer, as well as its possible magneto resistance.

1) The measurements are performed by applying a magnetic field in the perpendicular direction. When the magnetization switches direction, this is expected to give a change in the electronic transport in the magnetic (Ga,Mn) As. A well know example her is the anomalous Hall effect.

2) The authors describe their measurements as a "quasi four terminal "measurements. Since the junction resistance can be very low (about 100 Ohm)(see fig 2a) it can well be that any voltage drops (either longitudinal or transverse) which occur in the (Ga,Mn) As layer when the current flows in the plane can contribute to the measured ΔG . This is not discussed at all.

3) This is also important because it is emphasized in the literature (ref. 61 and references mentioned in there) that (for a two-terminal measurement) theoretically there cannot be a MR in linear response when the magnetization of one of the electrodes is (fully) reversed. For the case of a four-terminal measurement this IS possible, however the MR can be caused by other reasons than CISS related effects,

4) One way to check if there is a contribution of MR of the (Ga,Mn) As layer, is by changing the current path in this layer. Tthis can be done e.g. to change the connection of the current contact in fig 1b from the upper right to the lower left contact. There might be an additional background resistance, but this measurement will give valuable info about the possible role of the MR of the (Ga,Mn) As layer

5) The authors also show measurements on achiral molecules. They show similar effects. I think this again points to another (not CISS related) origin of the observed MR (see also 6).

6) In Figure S4 they also show the spin valve conductance of control junctions without any molecules, with both Al and Au electrodes. These ALSO show a "spin valve" resistance. This again seems to hint strongly to the fact that the MR this is caused by the magneto resistance of the (Ga, Mn) As layer. Note also that in this case the metallic top electrode will substantially short out any voltages in the in plane direction. Therefor the MR can be smaller than when molecules are used in the junction.

7) Starting at page 10 they give a theoretical description of their results. It is not clear to me if this is the same model as in ref 56. If so they should make an explicit remark about this.

8) It seems that this model gives a MR even in the linear transport regime. Can the authors explain this?

9) I am very surprised by the remark made on page 13 "and the CISS induced spin valve conductance is a high-order effect in the electron transport" But the fitting with the theory (in fig 5b) clearly shows that the theory also predicts a spin valve conductance for low current, in the linear transport regime. So it seems that there is no high order effect needed in the theory to obtain a finite MR at low bias.

Reviewer #2:

Remarks to the Author:

Interplay of Structural Chirality, Electron Spin and Topological Orbital in Chiral Molecular Spin Valves

The manuscript presents an interesting study on the importance of the contact, and spin orbit coupling in the contact, in explaining the spin selectivity efficiency in CISS based devices. The

explanation of this spin filtering high efficiency in CISS is under controversy. Many devices have shown very high spin polarization but usually utilize gold contact that have high spin orbit coupling. The manuscript shows that the contacts are important, and that CISS can become more efficient utilizing a gold contact. The heavy-metal electrode SOC is claimed provides an efficient way to convert the orbital polarization induced by the chiral molecular structure to spin polarization. The main results are achieved by directly comparing magnetoconductance measurements on spin valves with different metal electrodes. Using a spin valve with chiral molecules large differences between Au and Al electrodes are measure. Overall, the paper is clear to read, and the experiments are well made. The results supply significant information about the microscopic origin of the CISS effect, revealing indeed a fundamental relation between structure chirality, electron spin, and orbital.

The paper is therefore exciting and important, however there are some questions and clarifications should be answered before publication.

- 1) The adsorption of the monolayer would be different between devices. Since the authors don't use a tunnel barrier, large difference between devices could result from different pinholes geometry. The authors should supply in the SI a characterization of the adsorb layers and the devices (perhaps add TEM images of devices).
- 2) Adding to the first questions how many devices were measured for each electrode and molecule. The distribution of dG in these different cases should also be added to SI.
- 3) In that sense, please add error bars to figures 2,3,4.
- 4) The main claim of the paper is that the SOC in the metal electrode converts the orbital polarization into spin polarization. Yet it shows only a metal with large and small SOC. It would be good to present an intermediate effect with metals that have intermediate SOC. Adding Ag for example in this work or following work would be of great interest.
- 5) Using only very low temperatures could be a problem in studying CISS. CISS was found to be more effective in high temperatures. Can you show the temperature dependence of the effect measured here?
- 6) Why there is a large and different offset in the magnetoconductance in figure 2.
- 7) In Figure 4 we see large differences between the four molecules used on Al. Is this a noise issue? If not, what is different? The bottom contact is the same and you don't expect changes due to chirality according to the suggested model.
- 8) Why ODT is showing large dG both on Au and Al? Are the linkers the same?
- 9) In the theoretical model 6 fitting parameters are used. In that case one can expect a good fit. Can we relate the measured α 's to known parameters on Au and Al?

Two minor issues;

- 1) Please add the measured temperature on the plots.
- 2) In figure 2 a and b the green color doesn't match, please change

Response to Reviewer #1:

We thank the reviewer for the careful review and constructive criticisms. The reviewer raised several key experimental issues pertinent to the main conclusions of this work. While several of these issues were addressed in this manuscript and its SI or our previous work (*ACS Nano* 14, 15983 (2020)), it is apparent that further clarification and/or extended explanation are needed. Below we present detailed responses to the specific comments and list the corresponding changes made to the manuscript and SI.

0): The authors have performed spin valve magneto resistance measurements where they have assembled chiral and achiral molecules on magnetic (Ga, Mn)As. They observe a change in the MR modulation of the conductance when they compare Au and Al top electrodes. They interpret their results in terms of the role of the different strengths of the spin orbit interaction (SOC). They provide a theoretical model. I cannot recommend publication. The most important reason is that the authors observe MR even for achiral molecules, and even without any molecules at all. This strongly suggest that the MR does not originate from the molecular junction itself (due to CISS related effects), but comes from the magneto resistance (e.g. anomalous Hall effect, or related effects) of the (Ga,Mn) As layer. Despite the crucial role of this layer, (almost) nothing is mentioned about the characterization of the magnetic behavior this layer, as well as its possible magneto resistance.

Response: We will address the results of junctions with achiral molecules and more on control junctions without molecules later in response specifically to Comments 5 and 6. Here we first focus on the characterization of the (Ga,Mn)As and possible effect of its anomalous Hall effect (AHE). We were keenly aware of the crucial role of the (Ga,Mn)As layer, particularly its magnetotransport properties, on the correct interpretation of the junction magnetoresistance (MR) results. This was a critical first issue in our early work reported previously (*ACS Nano* 14, 15983 (2020)), where we devoted significant initial efforts on detailed characterization of the (Ga,Mn)As layer and ruling out spurious effects from its magnetoresistive responses. The results were discussed in the main text of our previous paper and elaborated in the SI. Here we reiterate the key results and direct attention to the relevant sections in the SI for details (<https://pubs.acs.org/doi/10.1021/acsnano.0c07438?goto=supporting-info>):

i) (Ga,Mn)As characterization: Despite the fact that the kind of perpendicularly magnetized epitaxial (Ga,Mn)As films have been exhaustively characterized by many groups over the years, in our experiments, we *always* employed a device structure shown schematically in Figure 2a and Figure S1a of the previous paper, in which the AHE and MR of the (Ga,Mn)As film can be measured on the *same* device that hosts the molecular junctions. This is so that we would always have precise knowledge of the properties of the (Ga,Mn)As in a particular device. A representative set of characterization results was presented in Section 1, Supporting Figure 1 of the SI entitled: “**Characterization of the (Ga,Mn)As thin films**” of the *ACS Nano* paper, which includes the AHE, and the temperature-dependent longitudinal and Hall resistivities.

- ii) Anomalous Hall effect: In Section 2, Supporting Note 2 of the SI of our *ACS Nano* paper, entitled: “**Ruling out the anomalous Hall effect (AHE) of the (Ga,Mn)As as the origin of the observed junction MR**”, we summarized a number of arguments against the AHE as the origin of the observed junction MR, based on the design and observations of our experiments. Further evidence against any significant contribution of the (Ga,Mn)As layer to the measured junction resistance includes the opposite temperature dependences of the film and junction resistances (Figures S1b and S1d versus Figure 2b of our previous paper). The experimental results we have collected since then have further strengthened the case, which we will present in detail below in response to Comments 1, 2 and 6.

We do appreciate the reviewer raising these points; it is apparent that they should be stressed again in the present manuscript. We have added a section in the current SI: "5. Potential contribution of AHE of (Ga,Mn)As in observed MC". A statement was added in the manuscript (page 6) to point to the SI section.

1): The measurements are performed by applying a magnetic field in the perpendicular direction. When the magnetization switches direction, this is expected to give a change in the electronic transport in the magnetic (Ga,Mn)As. A well-known example here is the anomalous Hall effect (AHE).

Response: As stated above, from the outset, we were always mindful of potential spurious contributions from the (Ga,Mn)As, especially the AHE, to the measured junction MR. By now, we have compiled a compelling case for ruling out the AHE as the origin of observed junction MR. Below we present an updated list of the main points in brevity.

- i) The junctions in our experiments were designed and fabricated as true vertical junctions (junction size ranging from $5 \times 5 \mu\text{m}^2$ to $15 \times 15 \mu\text{m}^2$) using lithography, in contrast to simple cross-stripe junctions. While such junctions are not four-terminal devices in the strictest sense, they are very close. Several observations have provided strong evidence that the contribution to the junction resistance due to current crowding in our devices is very small (more details below in response to Comment 2).
- ii) The MC of the molecular junctions show nontrivial dependences on the biases. Specifically, the MC of chiral molecular junctions show approximate linear dependence on the bias current over large ranges of biases. This is qualitatively different from the bias (in)dependence of the AHE of the (Ga,Mn)As: The Hall voltage is linear with the current, thus the Hall resistance or conductance is a constant, i.e., independent of the current.
- iii) If the AHE of (Ga,Mn)As actually contributed significantly to the junction MR, we would expect the effect to be more pronounced in junctions with lower resistances, especially control junctions without molecules (e.g., Fig. 3b of our previous paper) than in the molecular junctions. We now have performed systematic measurements in a large number of control samples with different NM electrodes (Au, Al, Ag, Cu); consistently, the control junctions showed no pronounced MC (more details below in response to Comment 6). Importantly, in

our experiments, the junction MC (ΔG) show no correlation with the total junction conductance; a detailed discussion was presented in Section 2 of the SI of the current manuscript, entitled: “*Spin valve conductance (ΔG) versus total junction conductance (G)*”.

- iv) An additional strong argument was prompted by Comment 4 of the reviewer (see detailed response below). If the junction MR was a spurious effect from the AHE, its sign would depend on the *relative* orientation of the bias current and applied magnetic field. In our experiments, *the sign of the molecular junction MR depends only on the field direction regardless of the measurement configuration*. This is strong evidence to exclude the role of AHE of the (Ga,Mn)As in our junction MR.

2): The authors describe their measurements as a “quasi four terminal” measurements. Since the junction resistance can be very low (about 100 Ohm) (see fig 2a) it can well be that any voltage drops (either longitudinal or transverse) which occur in the (Ga,Mn)As layer when the current flows in the plane can contribute to the measured delta G. This is not discussed at all.

Response: The effect of “current crowding” in planar junctions has been a well-known issue since their inception in early 1960s because of their intrinsic “quasi four-terminal” geometry. Indeed one must be certain that any voltage drops in either electrode do not contribute significantly to the measured junction resistance. We have high confidence that in our *molecular* junctions this is the case based on the following:

- i) In contrast to conventional doped semiconductors, the (Ga,Mn)As in our devices have unusually high doping levels of several *percent*. This results in carrier (hole) densities on the order of 10^{21} cm^{-3} and resistivities on the order of $\text{m}\Omega\text{-cm}$, close to those in metals.
- ii) The junction resistances of high-quality molecular junctions always show weakly insulating behavior, i.e. weakly increasing with decreasing temperature (Figure 2b of our previous paper). This is different from, even opposite of the temperature dependences of the (Ga,Mn)As film resistances, either longitudinal or transverse (Hall) (Figures S1b and S1d of our *ACS Nano* paper).
- iii) Also applicable here is point iii) in 1): Any current crowding effect is expected to be stronger in junctions with lower resistances, especially control junctions without molecules (e.g., Fig. 3b of our previous paper) than in the molecular junctions. This is clearly contrary to our experimental results.

We briefly mentioned some of these arguments in our previous paper. Again, we realized that it is a point that should be stressed in the current manuscript; we have incorporated these points into Section 5 of the revised SI of this manuscript.

3): This is also important because it is emphasized in the literature (ref. 61 and references mentioned in there) that (for a two-terminal measurement) theoretically there cannot be a MR in linear response when the magnetization of one of the electrodes is (fully) reversed. For the case of

a four-terminal measurement this IS possible, however the MR can be caused by other reasons than CISS related effects,

Response: Whether a two-terminal chiral molecular spin valve with a ferromagnetic electrode (spin analyzer) has a linear-response MR was a point of intense debate in the CISS field. It was the focal point of our previous work (*ACS Nano* 14, 15983 (2020)), where we clearly identified a nontrivial linear-response component in the spin valve conductance. By now, there are experimental observations from a number of groups unambiguously showing nonzero linear-response MR in 2T chiral molecular spin valves. A striking recent example is reported in *Nano Lett.* 22, 5022 (2022). Theoretically there is continuing debate on the issue, particularly on how to account for the experimental observations of MR in the linear regime. Several different scenarios have been suggested, including effort by the authors of Ref. 61 (*PRB* 104, 155420 (2021)). Our model of magneto-modulation of tunneling barrier offers a natural explanation for the observation (details are given in response to Comment 8 below). We respectfully disagree with the suggestion that the presence of linear-response MR necessarily implies the MR is caused by other reasons than CISS related effects.

4): One way to check if there is a contribution of MR of the (Ga,Mn)As layer, is by changing the current path in this layer. This can be done e.g. to change the connection of the current contact in fig 1b from the upper right to the lower left contact. There might be an additional background resistance, but this measurement will give valuable info about the possible role of the MR of the (Ga,Mn)As layer

Response: We thank the reviewers for the helpful suggestion. The particular measurement configuration depicted in Fig. 1b is a schematic representation only. Typically, for each device, before we commenced full sets of junction MR and I-V measurements, we would first perform a complete set of measurements (for one set of measurement conditions) using **all possible combinations of electrodes**. Consistently, all combinations of “quasi four-terminal” setups always yielded essentially the **same** results. We had overlooked the significance of this observation. As the reviewer correctly pointed out, this result in fact gives valuable information about possible spurious contribution from the (Ga,Mn)As layer: If the junction MR was a spurious contribution from the AHE in the (Ga,Mn)As layer, its sign would depend on the *relative orientation* of the bias current and applied magnetic field. Specifically, a change of current contacts suggested by the reviewer would lead to a sign change for the MR for the same field and current directions. In our experiments, *the sign of the molecular junction MR depends only on the field and current directions, regardless of the measurement configuration (current path)*.

We are grateful to the reviewer for the insight, which prompted us to identify this important point. We have added this to the SI, point iv) of Section 5.

5): The authors also show measurements on achiral molecules. They show similar effects. I think this again points to another (not CISS related) origin of the observed MR (see also 6).

Response: We clearly observed nonzero junction MR with achiral molecules. However, we disagree that this “points to another (not CISS related) origin of the observed MR”. In fact, a nontrivial MR in junctions of achiral, but inversion asymmetric, molecules is an explicit prediction of the orbital polarization model (*Nature Materials* 20, 638-644 (2021)). This issue was discussed in some detail in the original manuscript (page 10) in the section entitled: “**Chiral versus achiral molecules**”.

Both 16-mercaptohexadecanoic acid (MHA) and 1-octadecanethiol (ODT), the achiral molecules used in our study, are polar due to the different terminal groups in each, thus possess inherent inversion asymmetry. The molecular polarity (asymmetry) factors directly into the magnetochiral modulation of the tunneling barrier (Ref. 56), which is the mechanism for the spin-valve MR in our model (more on the model in responses to Comments 7 and 8). Physically, the magnetochiral modulation of the tunneling barrier originates from electrical magnetochiral anisotropy (eMChA) (page 11 of the manuscript and Ref. 56). It is therefore important to note that *eMChA has been definitively observed in both chiral and (nonchiral) polar structures* (e.g., the polar, nonchiral semiconductor BiTeI in Fig. 3 in *Nat. Commun.* 9:3740 (2018)). Quantitatively, the eMChA is stronger in a chiral crystal with well-defined structural chirality (e.g. the Te crystal in *PRB* 99, 245153 (2019)) than in a nonchiral polar solid. Therefore, our observations of finite spin-valve conductance in junctions of achiral (polar) molecules and their smaller magnitudes relative to that in chiral molecular junctions are fully consistent with the expectations of the theoretical model discussed in the manuscript and prior results of eMChA.

We have added a statement on page 10 and an extended discussion on page 13 in the manuscript to emphasize the importance of the polar nature of the achiral molecules and the relevance of prior results of eMChA in the chiral and polar systems. Three references (*PRL* 87, 236602 (2001), *Nat. Commun.* 9:3740 (2018), and *PRB* 99, 245153 (2019)) were added (Refs. 66,67 and 68 in updated manuscript).

6): In Figure S4 they also show the spin valve conductance of control junctions without any molecules, with both Al and Au electrodes. These ALSO show a “spin valve” resistance. This again seems to hint strongly to the fact that the MR this is caused by the magneto resistance of the (Ga, Mn) As layer. Also note that in this case the metallic top electrode will substantially short out any voltages in the in-plane direction. Therefore, the MR can be smaller than when molecules are used in the junction.

Response: Although the results of control junctions without molecules were presented and discussed in our previous paper (*ACS Nano* 14, 15983 (2020)), a more thorough examination of the issue is apparently needed. Moreover, we now have measured many more control samples with different NM electrodes (Au, Al, Ag and Cu) and are in position to place definitive upper bound on the MR in such junctions. The figure below shows a set of representative results, and we summarize the main points in the following.

i) The “spin-valve conductance” (ΔG) of the control junction shown in Figure S4 was extracted from the I-V measurements in opposite saturation fields; the original I-V data are shown in

panel (a) below. It is evident that the differences between the two are essentially at noise level, as seen in the resulting ΔG in panel (b) (red circles). The fluctuations and noise level were made even more apparent by a repeat measurement on the same junction (black squares). The repeat measurements also evidenced a lack of well-defined bias dependence for the ΔG . Furthermore, the values of ΔG determined from direct MR measurements were not consistent with those extracted from I-V measurements; in most cases, there were not well-defined MC curves (details in SI of our previous paper). These are in stark contrast with molecular junctions with Au electrode, where the I-V's, extracted ΔG , and their bias dependences are highly consistent, reproducible, and well-defined.

- ii) *Different junctions in different samples*: The minimal values and lack of consistent bias dependence of ΔG in the control junctions without molecules were also evidenced in comparison of different junctions in different control samples. Panel (c) shows ΔG of two control junctions with Au contacts. The values of ΔG are minimal without consistent bias dependence.
- iii) *Different NM electrodes*: Panel (d) shows a comparison of ΔG of control junctions with Au, Al, Cu and Ag electrodes. Again, in all cases, the values of ΔG are minimal without consistent bias dependence.

Taken together, the extensive measurements of control junctions without molecules established a solid case that the “spin-valve conductance” are minimal in magnitude and lacking any well-defined bias current dependence, which are qualitatively and quantitatively different from those in molecular junctions with Au electrode. Because of the serial nature of any contributions by the (Ga,Mn)As layer to the measured junction resistance, such contributions would manifest more strongly in the control junctions. The “spin-valve conductance” in the control junctions thus constitute an *upper bound* on any spurious effect from the (Ga,Mn)As layer, while the particular MC of the Au and Al control junctions shown in Fig. S4 were the highest among all our measurements of control junctions, thus representing a “worst-case scenario”.

The MC in the control samples therefore constitutes a viable reference for meaningful comparison of the MC in various molecular junctions. It is interesting to note that the molecular junctions with Au contacts show increasing MC for achiral and chiral molecules distinctly above the reference, whereas all molecular junctions with Al contacts show ΔG only slightly greater than the reference and no discernible differences between the achiral and chiral molecules. This lends further support to our conclusion that the NM electrode has a crucial role in the emergence of the spin-valve effect due to CISS in the molecular junctions.

We thank the reviewer for probing this issue. Prompted by the reviewer's comment, we have significantly revised and expanded Section 4 of the SI; key conclusions are emphasized in the main text (Page 10).

Fig. 1: Control samples. (a) Two separate measurements of I-V curves for a control junction (Au/(Ga,Mn)As) without any molecules in perpendicular magnetic fields of ± 700 Oe. (b) Bias dependence of ΔG extracted from the I-V curves in (a). (c) Comparison of bias-current dependent ΔG for two different control junctions with Au contacts from different samples. (d) Comparison of bias-current dependent ΔG of the control junctions with Au, Al, Cu and Ag electrodes.

7): Starting at page 10 they give a theoretical description of their results. It is not clear to me if this is the same model as in ref 56. If so, they should make an explicit remark about this.

Response: The analyses of the spin-valve conductance in this work were based on a modified version of the model in ref. 56. This was stated explicitly in the original manuscript (page 11) at the end of second paragraph in the section of “**Tunneling model and magnetochiral modulation of potential barrier**”: “We demonstrate that this model provides a semi-quantitative self-consistent description of all the key observations in the following.”. In the revised manuscript we have included ref. 56 in this sentence.

The modification of the model consists of incorporation of the magnetochiral modulation of the tunneling barrier into the Simmons model (Ref. 62 and 63 in the manuscript). Details of the modification and its application in the data analysis were presented in the following paragraphs of the section, and depicted in Fig. 5.

8): It seems that this model gives a MR even in the linear transport regime. Can the authors explain this?

Response: A remarkable feature of the (modified) model is that it offers natural explanations for all the key experimental observations summarized in the first paragraph of the section “**Tunneling model and magnetochiral modulation of potential barrier**”. In particular, it naturally yields a finite MR in the linear transport regime (zero bias). This is illustrated physically in Fig. 5a: The magnetochiral modulation leads to a *splitting of the tunneling barrier height* for the two magnetization directions of (Ga,Mn)As. This results in different junction conductances for the two magnetization states, i.e. finite spin-valve conductance, for all biases including the zero bias. This is demonstrated in numerical calculations shown in Fig. 5b.

We have expanded a statement in the “**Conclusions**” section (page 14) in the manuscript, summarizing the key experimental observations accounted for by the model.

9): I am very surprised by the remark made on page 13 “and the CISS induced spin valve conductance is a high-order effect in the electron transport” But the fitting with the theory (in fig 5b) clearly shows that the theory also predicts a spin valve conductance for low current, in the linear transport regime. So, it seems that there is no high order effect needed in the theory to obtain a finite MR at low bias.

Response: In theory, the Onsager’s reciprocity forbids a linear-response CISS-MR by consideration that the transport experiment is a perturbation of the equilibrium state. By “higher-order effect in the electron transport” we meant to state that, in our model, the CISS induced spin-valve conductance results from the nonequilibrium modulation of the tunneling barrier. Specifically, we propose that the current flow changes the tunnelling barrier, for example, to state A or B, depending on the magnetization and molecular chirality. Because the system keeps the memory of state A/B in the low current limit, it gives finite spin valve conductance in low current/ or linear regime. Thus, we circumvent a conflict with Onsager’s relation by involving highly nonequilibrium states. Because the conductance is exponentially sensitive to the tunneling barrier, the induced-spin valve conductance can be very large although the modulation (splitting) of the barrier may be small. In addition, Ref. 56 proposed interfacial charging as a potential scenario for the nonequilibrium states. Experimental investigation on their microscopic structure will be an interesting direction for future efforts. We agree with the reviewer that the term “higher-order” will likely cause confusion and may not be precise. We have revised the statement in question in the manuscript (page 13) so as to eliminate any potential confusion; we have removed the term “higher-order” and clarified the explanation.

Response to Reviewer #2:

Interplay of Structural Chirality, Electron Spin and Topological Orbital in Chiral Molecular Spin Valves

0) The manuscript presents an interesting study on the importance of the contact, and spin orbit coupling in the contact, in explaining the spin selectivity efficiency in CISS based devices. The explanation of this spin filtering high efficiency in CISS is under controversy. Many devices have shown very high spin polarization but usually utilize gold contact that have high spin orbit coupling. The manuscript shows that the contacts are important, and that CISS can become more efficient utilizing a gold contact. The heavy-metal electrode SOC is claimed provides an efficient way to convert the orbital polarization induced by the chiral molecular structure to spin polarization. The main results are achieved by directly comparing magnetoconductance measurements on spin valves with different metal electrodes. Using a spin valve with chiral molecules large differences between Au and Al electrodes are measure.

Overall, the paper is clear to read, and the experiments are well made. The results supply significant information about the microscopic origin of the CISS effect, revealing indeed a fundamental relation between structure chirality, electron spin, and orbital.

The paper is therefore exciting and important, however there are some questions and clarifications should be answered before publication.

Response: We thank the reviewer for the positive review and thoughtful comments on our manuscript. We particularly appreciate the reviewer's assessment that "the paper is exciting and important". The referee succinctly summarized the important results of our work. Below we address the specific questions/comments raised by the reviewer.

1). The adsorption of the monolayer would be different between devices. Since the authors don't use a tunnel barrier, large difference between devices could result from different pinholes geometry. The authors should supply in the SI a characterization of the adsorb layers and the devices (perhaps add TEM images of devices).

Response: This indeed is an important issue and we had previously expended significant effort on the characterization of SAMs of thiol molecules on both GaAs and (Ga,Mn)As surfaces (Ref. 57 and 58 in updated manuscript). The high quality of the SAMs was evidenced by detailed XPS measurements (Ref. 57), extensive lateral force microscopy (Refs. 57 and 58), and effectiveness of the SAMs as templates for directed self-assembly of Au nanoparticles (Ref. 58). The characterization of the presence and extent of pinholes in SAMs is much more challenging, even on Au surfaces. TEM is unlikely to provide useful information on this issue, due to its difficulty in imaging organic molecules and as importantly, limited field of view, thus incapable of discerning the molecular coverage.

That said, based on extensive experiments and analyses reported here and our previous paper (Ref. 53), we have high confidence that the magnetoconductance, ΔG , reflects exclusively transport

through the molecules and the presence of pinholes in the SAM is effectively mitigated by the use of a magnetic semiconductor (Ga,Mn)As. Although the direct contact contributes to the overall conductance of the junction, both the magnitude and bias current dependences of the CISS spin valve conductance (ΔG) of the junction is independent of electron transport through the direct contact. This issue was a focal point of discussion in our previous work (Ref. 53), and it was briefly mentioned in the current manuscript (2nd paragraph, page 6). Prompted by the reviewers' comments, we have significantly revised and expanded relevant discussion in the SI of the current manuscript. We summarize the main points in the following:

Sec. 2: Spin valve conductance (ΔG) versus total junction conductance (G). ΔG show no correlation with the total junction conductance, thus do not depend on the amount of pinholes in the SAM.

Sec. 4: Control samples without molecules. ΔG of the control junctions are essentially at noise level and show no well-defined bias dependence.

Sec. 5: Ruling out the role of AHE of (Ga,Mn)As in observed MC. The spin-valve conductance of the molecular junctions with Au contacts contain no significant contributions from the anomalous Hall effect in the (Ga,Mn)As film.

2) Adding to the first questions how many devices were measured for each electrode and molecule. The distribution of dG in these different cases should also be added to SI.

Response: In this work, for each combination of molecule and NM electrode, a full set of measurements was repeated in multiple samples (2 to 4), each with 4 junctions, and the results were consistent as summarized below.

Due to variations of the quality of molecular assembly (molecular coverage) in the junctions, there are substantial variations in the overall I-V characteristics and total junction conductance, and to a much lesser extent, in ΔG for the same molecule. The table below summarizes the distribution of ΔG of different molecular junctions with Au electrode near zero bias current and at higher bias currents. The range of magnitude of ΔG mentioned in table is taken from at least three junctions from different samples for each molecule.

Molecular junctions (Au electrode)	ΔG near zero bias current (μS)	ΔG at large bias current (μS)
AHPA-L	45-90	100-200
L-Cysteine	28-43	50-65
MHA	17-28	40-80
ODT	0-5	45-85

Table: The distribution of magnitude of ΔG of different molecular junctions with Au electrode near zero bias and at higher bias currents.

As we can see from the table, the distribution of ΔG for AHPA-L junctions approximately ranges from 45-90 μS to 100-200 μS . In the entire bias ranges, the ΔG is consistently much greater than

those of other molecules (L-Cysteine, MHA, and ODT). In case of junctions with Al contact, the distributions of ΔG show significant overlap for different molecules (see Fig. 4b and Fig. S5, and response to Comment 7 below), and are consistently lower than those of junctions with Au contact for any molecule.

3) In that sense, please add error bars to figures 2,3,4.

Response: As suggested by the reviewer, we have added error bars to figures 2,3, and 4.

The error bars are the standard deviations from the mean taken from multiple measurements of the IVs of each junction. So, each error bars reflect the noise levels in the measurements, not the range of ΔG variation among different samples.

4) The main claim of the paper is that the SOC in the metal electrode converts the orbital polarization into spin polarization. Yet it shows only a metal with large and small SOC. It would be good to present an intermediate effect with metals that have intermediate SOC. Adding Ag for example in this work or following work would be of great interest.

Response: We thank the reviewer for raising this point. Indeed, we have fabricated and measured large numbers (>10) of devices with Cu and Ag electrodes, two NM materials of intermediate SOC strengths between Au and Al. However, despite the repeated attempts, for either material, we were unable to obtain results with the degree of consistency seen in Au and Al devices. We speculate that this originated from poor interfaces of the molecular SAM with Cu and Ag, possibly due to the higher evaporation temperatures or some specific chemistry with the organic molecules for Cu and Ag.

The discussion of issue and some experimental results were presented Section 3 of the SI. Indeed these are experiments we continue to pursue with possibly “softer” techniques of putting down the top NM electrode.

5) Using only very low temperatures could be a problem in studying CISS. CISS was found to be more effective in high temperatures. Can you show the temperature dependence of the effect measured here?

Response: The temperature dependence of the CISS effect is a pertinent issue in the field. We had presented some results of the temperature dependence of MC in (Ga,Mn)As/AHPA-L/Au junctions in our previous work (Figure 3(a) of Ref. 53, *ACS Nano* 14, 15983 (2020)). We did observe a strong temperature dependence, however, it bears resemblance to that of the magnetization of (Ga,Mn)As. Because of this complication, we are unable to make any definitive statement regarding the temperature dependence of the CISS effect. This is another aspect we hope to obtain more definitive results in future experiments.

6) Why there is a large and different offset in the magnetoconductance in figure 2.

Response: We presume that the reviewer was referring to the “offsets” in Figs 2a and 2b. In our picture, the total junction conductance is the sum of parallel conduction through the molecular SAM and direct contact. Both contribute to the “offset”: The conductance of the direct contact is essentially field-independent; the conductance through the molecules is modulated by the magnetic

field (magnetization) which reflect the CISS effect and result in spin-valve conductance ΔG . However, ΔG is a relatively small modulation of the latter, which has a large field-independent background, contributing to the “offset”. This is best illustrated by Fig. 5a and the existence of a large α_o in our model analyses. The change of the “offset” with bias current is a direct reflection of the nonlinear nature of the I-V (bias-dependence of conductance).

In case the reviewer was referring to the “offset” in Fig. 2c, that “offset” is a finite zero-bias junction MC, namely a linear-response CISS spin valve effect. It was the central result and a focal point of discussion in our previous work (Ref. 53). Its origin and significance are discussed again in the current manuscript, and it is fully accounted for in the model of magnetochiral modulation of the tunneling barrier.

7) In Figure 4 we see large differences between the four molecules used on Al. Is this a noise issue? If not, what is different? The bottom contact is the same and you don’t expect changes due to chirality according to the suggested model.

Response: Of the four molecules used in the study, AHPA-L, MHA, and ODT yield similar MC in junctions with Al contact, and the values are only slightly above the reference level based on measurements on control junctions without any molecules (see the revised Section 4 of SI for details). The MC of the L-cysteine molecule junction seems to be even lower, which we speculate might be related to its much shorter molecular length. Therefore, the MC of the Al junctions are at or near the background/noise levels regardless of the type of molecules, thus are consistent with the expectation of the suggested model.

8) Why ODT is showing large dG both on Au and Al? Are the linkers the same?

Response: As discussed above, the Al junctions with ODT show MC similar to those of other molecules, and the values are at or near the background/noise levels. The Au junction with ODT does show a steeper *slope* than that of MHA and similar to that AHPA-L (importantly, the magnitude is still much smaller than those of AHPA-L junctions). We speculate that the differences between the ODT and MHA junctions are due to the different terminal groups: ODT and MHA both have a thiol end and similar carbon chain but with $-\text{CH}_3$ and $-\text{COOH}$ terminal groups respectively. The molecular asymmetry of ODT, with a nonpolar group at one end, should be stronger. We emphasize that we do not have a definitive explanation for this feature, but it has no impact on the key conclusions of the work.

9) In the theoretical model 6 fitting parameters are used. In that case one can expect a good fit. Can we relate the measured alpha’s to known parameters on Au and Al?

Response: The fitting was performed separately for positive and negative bias currents, using α_o , α_{-M} and α_M as fitting parameters. The value of β was kept constant at 10 V^{-1} while fitting bias dependence of both chiral and achiral molecules. So, there are 3, not 6, adjustable parameters in our fittings. The resulting values of α_{-M} and α_M offer a semi-quantitative measure of the CISS spin-valve conductance, which we surmise are connected to the strength of the SOC of the NM contact. The definitive and quantitative relations between the α_M values and specific parameters in the NM are an interesting open question which we hope will invite further theoretical work.

Two minor issues;

- 1) Please add the measured temperature on the plots.
- 2) In figure 2 a and b the green color doesn't match, please change

Response: We have labeled the measurement temperatures in all figures. We also changed the color in Figures 2a and 2b.

Reviewers' Comments:

Reviewer #1:

Remarks to the Author:

The authors have submitted an extensive reply to the previous reviewers's remarks, and made changes/additions to the manuscript. I have carefully studied this, and have come to the conclusion that this manuscript cannot be accepted for Nature Comm., for fundamental reasons. I give my argumentation below:

1) The overall behavior the authors observed is a MR (in particular Delta G) in the linear regime (for bias voltages $V < kT/e$), which increases by up to a factor of two at higher biases. Here it should be noted that the I/V characteristics can be nonlinear. This means that the manifestation of the MR at high bias can be a modification of the processes which (already) happen in the linear transport regime.

2) The observation of an MR in the linear transport regime is important, because to my knowledge in the linear regime there are no reliable theories which predict a MR (when the direction of the magnetization is fully reversed) for two-terminal measurements. The reference quoted by the authors (Phys. Rev. B 104, 1555420 (2021)) describes that there can be an MR in the linear regime when the direction of a magnetic field or magnetization is changed, but it vanishes for 180 degrees reversal, in agreement with the reciprocity theorem for two terminal measurements. From the experimental side, the experiments of Nano Lett. 22, 5022 (2022) do indeed seem to show that there is an MR in the linear regime under (full) magnetization reversal. As the authors point out this is indeed not understood

3) In response to the reviewers remarks they have repeated the experiments by changing the (current) contact configuration. It is not clear however if they tested the reciprocity theorem for four terminal measurements, which predicts that in the linear response regime the MR should change sign when current and voltage contacts are interchanged.

4) In other words, they have not convincingly shown that the results obtained in the linear transport regime are in contradiction with reciprocity. This still makes it possible that there is an effect of the magneto transport in the (Ga, Mn) AS layer.

5) They explain their results by fitting the data to a theory, which is described in detail in ref. 56. In that ref. 56 they use phrases like "...we propose that CISS is a higher order effect derived from magnetism and chirality dependent nonequilibrium states. CISS refrains from the Onsager's reciprocal relation because the charge accumulation breaks the microscopic reversibility" I cannot, and also will not (because this (yet unpublished) manuscript is not under review here) comment on the details of that theory, but it is clear that the authors describe the NON-LINEAR REGIME ONLY in which Onsager's reciprocity does not have to hold.

6) On page 11 of the current manuscript they have added "Here we propose that the current flow changes the tunneling barrier in a way that depends on the magnetization and molecular chirality". This is again a confirmation that the theory describes a modification of the transport parameters WHEN A BIAS IS APPLIED. It therefore cannot describe the linear transport regime, where the transport only depends on the equilibrium parameters of the system. They have also added: "Because the system retains memory of the non-equilibrium phase, it yields a finite spin valve conductance even at low current." This is totally incomprehensible. What is the "memory" the authors refer to? Why would the sample "remember" what is happening in the nonlinear non equilibrium regime, when it is measured in the linear regime?

They also write "Because the conductance is exponentially sensitive to the tunneling barrier, the induced spin valve conductance can be considerable even for a small modulation (splitting) of the barrier. This remark is in conflict with linear response. There is always a bias regime (of the order of kT/e), where the transport can be expressed in terms of the equilibrium properties of the device, and where current induced changes are not relevant.

7) Related to the above remarks I can absolutely not understand that the authors now use this theory to describe their data in BOTH linear and non-linear regime. For example, in Supplementary figure 6 they fit the bias dependence of the MR, and the fitting extends over the entire range of bias currents, including the low bias linear response regime. This is wrong because, as argued above, the authors theory cannot apply to the linear response regime

8) In the conclusions they have added " including finite linear regime magnetoresistance due to CISS in two-terminal molecular spin valves". Again this is wrong for the reasons above.

To conclude I cannot recommend publication of this manuscript because a theory which is designed explicitly to apply to the non-linear regime (see ref. 56 and my remarks above) is being

used also to explain the results in the linear regime. THIS IS NOT POSSIBLE! The authors theory simply has nothing to say about the linear transport regime. Therefore the observation of the MR in the linear regime is either a result of a) a yet not understood effect of the MR (and/or AHE) of the (Ga,Mn) As layer or b) a yet not understood (perhaps CISS related) effect which causes breaking of reciprocity in the linear transport regime. This would be a dramatic conclusion, since in the past decades reciprocity has been tested in all possible detail. There are only a few experiments (see also above) which seem to indicate that reciprocity may not work for CISS in the linear regime, but the consequences of this would be so dramatic that these results should be scrutinized in all detail. Finally, given the similar sizes of the observed MR in linear and nonlinear regimes, I therefore also doubt that the authors' theory (which in principle could indeed be valid in the non-linear regime) applies to the current experiments, even in the non-linear regime

Reviewer #2:

Remarks to the Author:

I think that the data is interesting and the authors answered in a good way both my and the first reviewer questions.

The measurements, and the interpretations do not supply the cleanest picture, nevertheless I feel that the results are good enough for publication.

I therefore recommend to publish the manuscript in Nature com.

Response to Reviewer #1:

Reply: We thank Reviewer #1 for the review. We note that in the first report, Reviewer #1 focused on a range of experimental objections to our work. In our response and revision, we addressed those objections in depth, and the reviewer did not make further comments on those issues. Instead, the reviewer's objection now centered on our observation of near zero-bias two-terminal CISS spin valve effect, based on the expectation from the Onsager reciprocity.

Comments:

The authors have submitted an extensive reply to the previous reviewers's remarks, and made changes/additions to the manuscript. I have carefully studied this, and have come to the conclusion that this manuscript cannot be accepted for Nature Comm., for fundamental reasons. I give my argumentation below:

1) The overall behavior the authors observed is a MR (in particular Delta G) in the linear regime (for bias voltages $V < kT/e$), which increases by up to a factor of two at higher biases. Here it should be noted that the I/V characteristics can be nonlinear. This means that the manifestation of the MR at high bias can be a modification of the processes which (already) happen in the linear transport regime.

2) The observation of an MR in the linear transport regime is important, because to my knowledge in the linear regime there are no reliable theories which predict a MR (when the direction of the magnetization is fully reversed) for two-terminal measurements. The reference quoted by the authors (Phys. Rev. B 104, 1555420 (2021)) describes that there can be an MR in the linear regime when the direction of a magnetic field or magnetization is changed, but it vanishes for 180 degrees reversal, in agreement with the reciprocity theorem for two terminal measurements. From the experimental side, the experiments of Nano Lett. 22, 5022 (2022) do indeed seem to show that there is an MR in the linear regime under (full) magnetization reversal. As the authors point out this is indeed not understood.

3) In response to the reviewers remarks they have repeated the experiments by changing the (current) contact configuration. It is not clear however if they tested the reciprocity theorem for four terminal measurements, which predicts that in the linear response regime the MR should change sign when current and voltage contacts are interchanged.

4) In other words, they have not convincingly shown that the results obtained in the linear transport regime are in contradiction with reciprocity. This still makes it possible that there is an effect of the magneto transport in the (Ga, Mn) As layer.

5) They explain their results by fitting the data to a theory, which is described in detail in ref. 56. In that ref. 56 they use phrases like "...we propose that CISS is a higher order effect derived from magnetism and chirality dependent nonequilibrium states. CISS refrains from the Onsager's reciprocal relation because the charge accumulation breaks the microscopic reversibility" I cannot, and also will not (because this (yet unpublished) manuscript is not under review here) comment on the details of that theory, but it is clear that the authors describe the NON-LINEAR REGIME ONLY in which Onsager's reciprocity does not have to hold.

7) Related to the above remarks I can absolutely not understand that the authors now use this theory to describe their data in BOTH linear and non-linear regime. For example, in Supplementary figure 6 they fit the bias dependence of the MR, and the fitting extends over the entire range of bias currents, including the low bias linear response regime. This is wrong because, as argued above, the authors theory cannot apply to the linear response regime.

8) In the conclusions they have added “including finite linear regime magnetoresistance due to CISS in two-terminal molecular spin valves”. Again, this is wrong for the reasons above.

To conclude I cannot recommend publication of this manuscript because a theory which is designed explicitly to apply to the non-linear regime (see ref. 56 and my remarks above) is being used also to explain the results in the linear regime. THIS IS NOT POSSIBLE! The authors theory simply has nothing to say about the linear transport regime. Therefore the observation of the MR in the linear regime is either a result of a) a yet not understood effect of the MR (and/or AHE) of the (Ga,Mn) As layer or b) a yet not understood (perhaps CISS related) effect which causes breaking of reciprocity in the linear transport regime. This would be a dramatic conclusion, since in the past decades reciprocity has been tested in all possible detail. There are only a few experiments (see also above) which seem to indicate that reciprocity may not work for CISS in the linear regime, but the consequences of this would be so dramatic that these results should be scrutinized in all detail. Finally, given the similar sizes of the observed MR in linear and nonlinear regimes, I therefore also doubt that the authors’ theory (which in principle could indeed be valid in the non-linear regime) applies to the current experiments, even in the non-linear regime

Reply: Because the reviewer’s comments are essentially based on a single argument: no linear-response MR should be observed in two-terminal devices because of the Onsager reciprocity, we answer them together in the following.

We first point out that the primary focus of this manuscript is on the elucidation of the role of electrode SOC in the CISS effect. The point raised by the reviewer, on the validity/interpretation of the transport results at small biases by invoking the Onsager reciprocal relation, has been the topic of many theoretical and experimental works in the past five years. A prior work of ours (ACS Nano 14, 15983 (2020)), a predecessor of this work, specifically targeted this issue. The reviewer’s comments and some of the theoretical works are based on the same premise --- The transport should be in the linear-response region at zero bias limit so that the Onsager relation must hold at zero bias limit, i.e., $G(M) = G(-M)$ at $V \rightarrow 0$.

First and foremost, since 2011, experimentally, there are now a large number of CISS magnetoresistance measurements in two-terminal devices. Wherever there is a clear delineation of the low-bias MR, the results consistently point to $G(M) \neq G(-M)$ at $V \rightarrow 0$, i.e., Onsager relation does not appear to hold even at $V=0$. Below is an incomplete list of such experiments, where one can find distinct finite changes of $G=dI/dV$ when flipping M or B at $V \rightarrow 0$ or $V=0$.

1. Figure 2, Kiran *et al.*, Adv. Mater. 28, 1957–1962 (2016)

2. Figure 4, Liu *et al.*, ACS Nano 14, 15983–15991 (2020)

3. Figure 2, Kim *et al.*, Science 371, 1129–1133 (2021)

4. Figure 4, Qian *et al.*, Nature 606, 902–908 (2022)

5. Figure 4, Al-Bustami *et al.*, *Nano Lett.* 22, 5022–5028 (2022)

6. Figure 2, Kulkarni *et al.*, *Adv. Mater.* 32, 1904965 (2020).

These devices covered a variety of chiral molecules and several different ferromagnets. “a yet not understood effect of the MR (and/or AHE) of the (Ga,Mn)As layer” undoubtedly cannot account for all these observations.

We are familiar with and have followed closely the theoretical controversy surrounding the Onsager relation discussed in [Phys. Rev. B 104, 155420 (2021)] and earlier relevant literature, e.g., Nano Letters 20, 6148–6154 (2020), Phys. Rev. B 99, 024418 (2019). There have been discussions, for example in a comment article [PRB 101, 026403 (2020)] to the latter paper [PRB 99, 024418(2019)], that CISS transport may not necessarily be constrained by the Onsager’s relation in the molecular devices. On the other hand, in the past ten years, transport *experiments* in two-terminal CISS devices have consistently reported finite MR even near zero bias limit (where MR was clearly measured near zero bias). This manuscript, notwithstanding its separate primary focus, adds to the preponderance of the experimental results. We emphasize that the experimental results do **not** invalidate the fundamental Onsager relation, rather, they raise the question of whether it is applicable to the molecular devices. This is an active and fast developing topic; to our knowledge, several theory groups are actively exploring viable alternative mechanisms. We therefore disagree with the rationale that an experimental work should be rejected for publication because an aspect of its results contradicts one’s theoretical expectation. Below we address the specific criticisms of our tunneling model.

Regarding the reviewer’s comment on the relative magnitudes of linear and nonlinear components of the CISS spin valve conductance, we point out that this comparison is not meaningful because of the likely presence of parallel conduction through direct contact in the junctions. As a case in point, in a junction exhibiting more nonlinear I-V and smaller contribution from direct contact, the nonlinear component is almost 10 times that of the linear component (Figure 4, ACS Nano 14, 15983–15991 (2020)).

Comment:

6) On page 11 of the current manuscript they have added “Here we propose that the current flow changes the tunneling barrier in a way that depends on the magnetization and molecular chirality”. This is again a confirmation that the theory describes a modification of the transport parameters WHEN A BIAS IS APPLIED. It therefore cannot describe the linear transport regime, where the transport only depends on the equilibrium parameters of the system. They have also added: “Because the system retains memory of the non-equilibrium phase, it yields a finite spin valve conductance even at low current.” This is totally incomprehensible. What is the “memory” the authors refer to? Why would the sample “remember” what is happening in the nonlinear non equilibrium regime, when it is measured in the linear regime?

They also write “Because the conductance is exponentially sensitive to the tunneling barrier, the induced spin valve conductance can be considerable even for a small modulation (splitting) of the barrier. This remark is in conflict with linear response. There is always a bias regime (of the order of kT/e), where the transport can be expressed in terms of the equilibrium properties of the device, and where current induced changes are not relevant.

Response: We reiterate that the main scope of this work is to examine the SOC origin from electrodes in CISS, for which the reviewer made no comment about in the second report. To account for the observed varying magnitudes of the spin valve conductance, we proposed a barrier modulation model which describes very well our data of both the variation and bias dependence. This is still a phenomenological model. As a possible physical origin of the model, we surmise that the barrier modulation originates from a charging effect in the device, and the “memory” refers to the hysteresis in the charging/discharging process possibly due to charge trapping. Although we do not have direct experimental visualization of the dynamic local charge distribution with current flow in the device, hysteresis in electric charging/polarization is quite common. Furthermore, the exponential sensitivity of the conductance to the tunneling barrier (modulation) makes any small charging hysteresis readily experimentally observable; conversely, it makes the theoretical linear bias regime (of the order of kT/e) exponentially more inaccessible, especially at our measurement temperatures. While still a proposition, this scenario provides a self-consistent account of our experiments. We anticipate that the proposed model will be tested by further experiments and stimulate further theoretical studies, leading to deeper understanding of the CISS effect.

To avoid potential confusion, we rephrase the “memory” effect in the main text (page 11) as:

“Here, we propose that a current flow changes the tunnelling barrier in a way that depends on the magnetization and molecular chirality. In this scenario, the barrier is modified by the induced charge accumulation. Any charge trapping in the device with changing bias would produce a hysteresis in the charging/discharging process, thus yielding finite spin valve conductance even at low currents and the system circumvents the Onsager’s reciprocal relation.”

In the “Conclusions” section (page 14), we changed “finite linear-regime magnetoresistance” to “finite low-bias magnetoresistance”, to more accurately reflect the proposed picture.

Reviewer #2 (Remarks to the Author):

Comments:

I think that the data is interesting and the authors answered in a good way both my and the first reviewer questions.

The measurements, and the interpretations do not supply the cleanest picture, nevertheless I feel that the results are good enough for publication.

I therefore recommend to publish the manuscript in Nature com.

Response: We thank Reviewer #2 for his/her review and support for the publication of our work.

Reviewers' Comments:

Reviewer #3:

Remarks to the Author:

The paper "Interplay of Structural Chirality, Electron Spin, and Topological Orbital in Chiral Molecular Spin Valves" by Y. Adhikari discusses the chirality-induced spin selectivity (CISS) and presents experimental evidence of the induced spin-filtering mechanism in Refs. [39, 49].

I have read the rebuttal letter and the revised manuscript. The previous reviewer criticized the authors' experiment, as it demonstrates finite magneto-conductance in the linear response regime, which contradicts the Onsager relation. Overall, I share the same opinion as the previous reviewer. The paper gives the impression that their measurements violate one of the fundamental laws of physics.

The authors fit their experimental data by adopting a previous theory arXiv:2201.03623 (2022), Ref. [56]. In this theory, the modification of the tunnel barrier by a possible bistable "charge trap" is introduced to explain the large magneto-conductance. Since the Onsager relation is valid close to the equilibrium regime, if the measurement has been done in a timescale shorter than the equilibration time of the bistable charge trap, the Onsager relation is not applicable. I agree with the authors that if the state is trapped at one of the energy minima, the finite linear magneto-conductance may be observed.

One way to check the authors' claim is to perform current measurements in a timescale sufficiently longer than the equilibration time (the timescale determined by the Arrhenius law) of the charge trapping states. I think this additional measurement would confirm the Onsager relation. If this experiment is difficult, the authors can show experimentally the hysteresis in the current-voltage characteristic, which would prove the existence of the charge trapping states. I think it is possible since the authors mentioned in the text that 'Any charge trapping in the device with charging bias would produce hysteresis in the charging/discharging process'.

To avoid giving the impression that the experiments violate fundamental laws of physics, the authors should address the residual magnetoresistance in the linear response regime more explicitly. A paragraph dedicated to this topic would be beneficial. Additionally, the authors could provide a list of several experiments that exhibit linear response magneto-conductance. To further support their claims, it would be even better if the authors could refer to a few papers that discuss the slow dynamics and the violation of the Onsager relation or the fluctuation dissipation theorem.

The magneto-conductance in the linear response regime should likely be a critically discussed issue in the CISS community, as the authors have pointed out. I am uncertain whether the induced spin-filtering mechanism, combined with Ref. [56], can quantitatively recover the fitting parameters presented in Table 1. Despite several ambiguities, I believe the experiment presented in the paper is interesting and worth publishing in some form eventually.

Reviewer #4:

Remarks to the Author:

The paper is presenting a very detailed study of magnetoconductance in a setup consisting of a magnetic (Ga,Mn)As electrode with out-of-plane magnetization and Au (or Al) electrodes with a SAM of chiral molecules in-between. The authors rationalize their results in terms of a conversion of orbital polarization induced by the chiral molecular frame to spin polarization due to SOC in the heavy-metal electrode. While the experiments seem to me quite solid, I have some comments and questions

-- I do not understand very well the proposed model in Eq. 2: if it is aiming at describing something like a magnetochiral anisotropy effect, then the conductance should contain something like a term proportional to $B \cdot I$, right? Is this the case in Eq. 2? Or are the authors referring to a non-linear bias-dependence? Moreover, why is the conductance increasing exponentially with bias? Simmons model yields roughly $I \sim \exp(-A \cdot V)$.

-- Why are the currents for the case of AHPA with a length of more than 5 nm still in the micro-amp range (I would expect currents in nanoAmp range). Are there still direct transport pathways between the electrodes? I also did not understand why there is no gap in the IV curves in the Supp Inf, while a clear gap is visible in similar experiments cited by the authors in Supp Fig. 2a using the same molecules.

-- How does the theoretical model cope with the very recent results at Pramanik's group, where CISS signals were found with magnetic fields *perpendicular* to the current paths and which seem to contradict the main idea of MCA as a source of CISS, where the scalar product between B and the current I would imply a zero contribution from this effect?
[<https://doi.org/10.1039/D2NH00502F>]

-- I agree with the Reviewer 1 that the model proposed in Ref. 56 seems to be addressing the non-linear bias regime, since it explicitly goes to second order in the applied bias (otherwise there is no MCA). Thus, it is a little confusing for me that MCA is invoked as a rationale for CISS also in the low bias regime. Of course, a fit can be done over the whole bias range, but in the linear regime no MCA can be used to explain anything.

Concerning the Onsager relation, I must admit that I do not have a clear opinion at the moment. As the authors correctly indicate in their reply, there are few experiments suggesting a non-zero polarization in the linear response regime down to zero bias, which apparently suggest a violation of Onsager in 2 terminal setups, but I do not see currently any explanation of it. I think that inelastic dephasing effects and the asymmetric 2-terminal geometry may be the source of the breaking of Onsager, but this is speculative. As the Reviewer suggests, a 4-terminal measurement could contribute to shed light on this issue.

A minor correction:

In Fig. 2c I have the impression that the color code is not correct

"Pink squares are measured from I-V curves, whereas black circles are measured from MC measurements at different bias currents." For me the squares look grey and orange (and filled), the empty circles look blue-ish

As a result, although the experimental part seems to me very well and accurately performed, I share the concerns of the Reviewer on the modelling part and on the rationalization of the experimental results. I can, therefore, not suggest the paper for publication in its present form

Response to Reviewer #3:

0): The paper "Interplay of Structural Chirality, Electron Spin, and Topological Orbital in Chiral Molecular Spin Valves" by Y. Adhikari discusses the chirality-induced spin selectivity (CISS) and presents experimental evidence of the induced spin-filtering mechanism in Refs. [39, 49].

I have read the rebuttal letter and the revised manuscript. The previous reviewer criticized the authors' experiment, as it demonstrates finite magneto-conductance in the linear response regime, which contradicts the Onsager relation. Overall, I share the same opinion as the previous reviewer. The paper gives the impression that their measurements violate one of the fundamental laws of physics.

Response: We thank the reviewer for the careful review of our manuscript and the thoughtful suggestions. The reviewer was particularly concerned about the observation of finite magneto-conductance in the linear-response regime, which seemingly contradicts the Onsager relation. It is certainly not our intention to suggest or imply that the observation of low-bias ('linear-response') magnetoconductance in the two-terminal devices, by us or the many others, is evidence for the violation of the Onsager relation. As stated in our responses to the previous reviewer, the point in question, on the transport results at small biases and their relevance to the Onsager relation, has been the topic of many theoretical and experimental works recently. A prior work of ours (ACS Nano 14, 15983 (2020)), a predecessor of this work, specifically targeted this issue. In spite of the different focus of the present manuscript, it seems that a further clarification/discussion of this issue is needed. Taking the reviewer's suggestions, we have added a paragraph dedicated to discussion of this topic in the revised manuscript (page 14). Below we present detailed responses to the specific comments and a list of the points made in the added paragraph.

1): The authors fit their experimental data by adopting a previous theory arXiv:2201.03623 (2022), Ref. [56]. In this theory, the modification of the tunnel barrier by a possible bistable "charge trap" is introduced to explain the large magneto-conductance. Since the Onsager relation is valid close to the equilibrium regime, if the measurement has been done in a timescale shorter than the equilibration time of the bistable charge trap, the Onsager relation is not applicable. I agree with the authors that if the state is trapped at one of the energy minima, the finite linear magneto-conductance may be observed.

One way to check the authors' claim is to perform current measurements in a timescale sufficiently longer than the equilibration time (the timescale determined by the Arrhenius law) of the charge trapping states. I think this additional measurement would confirm the Onsager relation. If this experiment is difficult, the authors can show experimentally the hysteresis in the current-voltage characteristic, which would prove the existence of the charge trapping states. I think it is possible since the authors mentioned in the text that 'Any charge trapping in the device with charging bias would produce hysteresis in the charging/discharging process'.

Response: We appreciate the specific suggestions offered by the reviewer. Regarding the first suggestion, in our experiments, the magnetoconductance (MC) measurements were performed at various fixed currents by sweeping the applied magnetic field (e.g., Figure 2a). Each MC loop typically took about 6 to 8 minutes. Within this time scale, we did not observe any measurable

changes in the MC. This implies that the “equilibration time” is significantly shorter or longer than the measurement time. Considering our measurement temperature (~ 5 K) and the stability of the observed I-V hysteresis in some molecular junctions at higher temperatures (see, for example, *Nat. Mat.* 5, 64 (2006) and more discussions below), the latter is far more likely. Therefore, the suggested experiment to observe any time dependence of the MC would likely require impractically long measurements, if at all possible.

As for possible hysteresis in the I-V characteristics, we first note that bias-induced conductance switching and I-V hysteresis have been observed in a variety of molecular junctions of different configurations (single molecule and self-assembled monolayer) and different molecules (with and without metal complexes). A few representative papers (r1-r4) and a review (r5) are listed below:

- r1. J. He *et al.*, Nat. Mat. 5, 64 (2006)
- r2. C.A. Nijhuis *et al.*, JACS 131, 17814 (2009)
- r3. F. Schwarz *et al.*, Nat. Nano 11, 170 (2016)
- r4. J. Park *et al.*, ACS Nano 16, 3, 4206 (2022)
- r5. G. Kastlunger and R. Stadler, Monatsh Chem. 147, 1675 (2016)

A couple of observations are especially relevant here: i) In experiments where the I-V hysteresis was carefully measured at low biases, it was clear that the hysteresis extends to the low-bias regime. ii) The observed charging dynamics show various time constants. In some cases, the bias-induced changes were essentially persistent, showing no relaxation in days, even at room temperature.

In our experiments, the majority of the molecular junctions were measured without particular regard to the bias history where no bias-induced changes in I-V and MC were clearly identified. However, in one of the devices where particular care was taken to examine the effect of bias history, pronounced bias-induced changes in I-V and MC were observed. Part of the results was presented in Supporting Figure 5 (Fig. S5) for our prior work (*ACS Nano* 14, 15983 (2020), <https://pubs.acs.org/doi/10.1021/acsnano.0c07438?goto=supporting-info>). A most striking feature was that for applied bias currents up to 800 μ A, the junction showed negligible low-bias (linear-response) MC, whereas the application of a 1000 μ A bias current induced substantial increases in MC over the full bias range; most notably, a significant ‘linear-response’ MC was now present. Concurrently, the I-V characteristics of the junction also saw significant changes due to the applied large bias. The induced changes in I-V and MC were essentially permanent, showing no observable relaxation over the experimental time scale (days) and conditions (kept at cryogenic temperatures).

Obviously, this result predated the model we are proposing here, thus no connection was made at the time between the observation and the Onsager relation. We thank the reviewer for prompting us to make the connection. We have included the results of bias-induced I-V and MC changes in the Supplementary Information (Section 8).

2): To avoid giving the impression that the experiments violate fundamental laws of physics, the authors should address the residual magnetoresistance in the linear response regime more explicitly. A paragraph dedicated to this topic would be beneficial. Additionally, the authors could provide a list of several experiments that exhibit linear response magneto-conductance. To further

support their claims, it would be even better if the authors could refer to a few papers that discuss the slow dynamics and the violation of the Onsager relation or the fluctuation dissipation theorem.

Response: Taking the reviewer's suggestion, we have added a paragraph addressing the observation of 'linear-response' magnetoconductance in the manuscript (page 14). We demonstrate the ubiquity of observation by referring to an incomplete list of experiments showing the effect, including one just appeared online in *Nature Chemistry* (Fig. 3c, r10 below). The ubiquity of the effect in such diverse variety of molecular systems and device structures argues strongly for the experimental veracity of the low-bias MC; the observations do not imply any violation of fundamental laws of physics, but rather present a theoretical challenge of reconciliation with the Onsager relation.

- r6. Kulkarni *et al.*, Adv. Mater. 32, 1904965 (2020). (Figure 2)
- r7. Liu *et al.*, ACS Nano 14, 15983–15991 (2020). (Figures 2,4,5)
- r8. Qian *et al.*, Nature 606, 902–908 (2022). (extended data Fig. 6)
- r9. Al-Bustami *et al.*, Nano Lett. 22, 5022–5028 (2022). (Figure 4)
- r10. Yang *et al.*, <https://doi.org/10.1038/s41557-023-01212-2> (Figure 3c)

In the previous version of the manuscript, we had suggested a possible way out of the apparent conundrum based on the model of Ref. [56] and a conjecture of hysteretic charging process. Taking the reviewer's advice, we have now included a list of references on the charge dynamics and bias-induced conductance switching and I-V hysteresis in molecular junctions (r1-r5 above). We note that slow electron dynamics has been reported in a variety of insulating solid-state and molecular materials over decades. Here we refer to the experiments in molecular junctions, which are more closely related. In the paragraph, we also refer to the bias-induced changes in MC and I-V in one of our devices, which are now included in the Supplementary Information (Section 8).

Finally, we are keenly aware that although the proposed scenario by us provides a self-consistent account of some of the most prominent (and most debated) observations in CISS spin valves (as summarized in the Conclusions section), it is one possible explanation for the experimental observations and certainly not definitive. With that in mind, we have taken care in the revised manuscript to convey that the model analysis and the resulting conclusions are not a final resolution of the issue.

3): The magneto-conductance in the linear response regime should likely be a critically discussed issue in the CISS community, as the authors have pointed out. I am uncertain whether the induced spin-filtering mechanism, combined with Ref. [56], can quantitatively recover the fitting parameters presented in Table 1. Despite several ambiguities, I believe the experiment presented in the paper is interesting and worth publishing in some form eventually.

Response: Indeed, the issue of linear-response magnetoconductance in two-terminal chiral spin valves has been vigorously debated in the CISS community. While its theoretical interpretation continues to elicit controversy and await a definitive resolution, there is increasing support for the veracity of the experimental observations as elaborated above. We appreciate the reviewer's positive assessment of our experiment, and we believe its publication will contribute to the final resolution of this important controversial issue.

Response to Reviewer #4:

0): The paper is presenting a very detailed study of magnetoconductance in a setup consisting of a magnetic (Ga,Mn)As electrode with out-of-plane magnetization and Au (or Al) electrodes with a SAM of chiral molecules in-between. The authors rationalize their results in terms of a conversion of orbital polarization induced by the chiral molecular frame to spin polarization due to SOC in the heavy-metal electrode. While the experiments seem to me quite solid, I have some comments and questions.

Response: We appreciate the reviewer's positive feedback on our manuscript, especially the positive assessment of the accuracy and significance of our experiments. We value the reviewer's comments and questions; we address each point below, which we believe has further enhanced the clarity of our work.

1): I do not understand very well the proposed model in Eq. 2: if it is aiming at describing something like a magnetochiral anisotropy effect, then the conductance should contain something like a term proportional to $B \cdot I$, right? Is this the case in Eq. 2? Or are the authors referring to a non-linear bias-dependence? Moreover, why is the conductance increasing exponentially with bias? Simmons model yields roughly $I \sim \exp(-A \cdot V)$.

Response: The proposed model presented in the manuscript is based on a combination of the magnetochiral modulation of the potential barrier (Ref. [56]) and the Simmons tunneling model. The essence of the model is depicted in Fig. 5a. Most importantly, the magnetochiral effect here is NOT the electrical magnetochiral anisotropy (eMChA) of the conductance of a chiral *conductor*, but rather magnetochiral modulation of the tunneling barrier across the *insulating* chiral molecule. In addition, we stress that eMChA respects the Onsager relation and does not agree with the I-V characteristics observed in our work and many other CISS experiments (e.g., r5-r10 in response to Reviewer 3).

Namely, in the molecular junctions, the $B \cdot I$ term does not directly go into the junction conductance as in eMChA, but rather manifests as a magnetization-dependent splitting of the tunneling potential barrier (Fig. 5a). The MC thus results overwhelmingly from the barrier splitting (exponential) rather than eMChA directly (linear with I). This is the essential physical origin of Eq. 2 within the proposed model: The molecular chirality and electrode magnetism combine to modulate the tunneling barrier through the insulating molecular junction, resulting in the change in junction conductance upon switching the magnetization. The exponentially increasing (differential) conductance with bias in Eq. 2 results readily from the approximate exponential I-V noted by the reviewer, which follows from an approximation of the Simmons model.

2): Why are the currents for the case of AHPA with a length of more than 5 nm still in the micro-amp range (I would expect currents in nanoAmp range). Are there still direct transport pathways between the electrodes? I also did not understand why there is no gap in the IV curves in the Supp Inf, while a clear gap is visible in similar experiments cited by the authors in Supp Fig. 2a using the same molecules.

Response: We appreciate the reviewer's astute observations. Several factors contribute to the large currents in our AHPA junctions. The most important factor, as correctly noted by the reviewer, is the presence of direct transport pathways between the electrodes. The self-assembled monolayer (SAM) is most likely not perfect at our device length scales (μm s), defects are almost always present, resulting in direct contacts between the NM and (Ga,Mn)As. This was the reason why we use the quantity ΔG as the accurate measure of the CISS spin valve conductance of the junctions. This issue was a focal point of discussion in our previous work (Ref. 53), and it is discussed again in the current manuscript (2nd paragraph, page 6) and SI (1 and 2). Another contributing factor is that the molecules in the AHPA SAMs are known to have a large tilt angle away from normal (see, for example, Ref. 53, the paragraph below Fig. 2), thus the 'effective' molecular length may be significantly shorter.

Varying extent of direct contacts in different junctions also accounts for the varying degree of nonlinearity in their I-V's. The junction in Supplementary Fig. 2a contains the least contribution from transport through the direct contacts, and the I-V shows the strongest nonlinearity (gap-like feature) and best reflects the I-V characteristics of the AHPA molecules. Correspondingly, this junction exhibits the largest ΔG values and steepest slope of increasing ΔG with bias current. These points are presented in Supplementary Information 2 (Spin valve conductance (ΔG) versus total junction conductance (G)).

3): How does the theoretical model cope with the very recent results at Pramanik's group, where CISS signals were found with magnetic fields *perpendicular* to the current paths and which seem to contradict the main idea of MCA as a source of CISS, where the scalar product between B and the current I would imply a zero contribution from this effect? [<https://doi.org/10.1039/D2NH00502F>]

Response: We thank the reviewer for bringing this recent paper to our attention. As explained in the response to point 1) above, the proposed model makes a clear distinction between the eMChA in a chiral *conductor* and magnetochiral modulation of the tunneling barrier across the *insulating* chiral molecules. The referenced work, which measures diffusive transport over μm length scales, is an excellent example for the former, in contrast to tunneling transport across insulating molecules in our devices. In fact, the transverse MC was recognized as a type of eMChA by the Rikken et al. (who discovered the longitudinal MC), for example, see Eq. 6 of PRL 94, 016601 (2005). We have added this point in the revised manuscript (bottom of page 14) and included this paper in the references.

4): I agree with Reviewer 1 that the model proposed in Ref. 56 seems to be addressing the non-linear bias regime, since it explicitly goes to second order in the applied bias (otherwise there is no MCA). Thus, it is a little confusing for me that MCA is invoked as a rationale for CISS also in the low bias regime. Of course, a fit can be done over the whole bias range, but in the linear regime no MCA can be used to explain anything.

Response: We have addressed this point in depth in responses to comments by Reviewer 3. Briefly, our model presents a possible explanation for the finite MC in the low-bias ('linear-response') regime based on MCA modulation of the tunneling barrier and possible charging hysteresis (bias-

induced I-V and conductance changes) in the molecular junctions. We emphasize that the proposed model provides a self-consistent account of the key experimental observations including the magnitude, sign, and bias dependence of the CISS spin-valve MC; it is not meant to be a definitive theory and we have taken the care in the revised manuscript to make that clear.

5): Concerning the Onsager relation, I must admit that I do not have a clear opinion at the moment. As the authors correctly indicate in their reply, there are few experiments suggesting a non-zero polarization in the linear response regime down to zero bias, which apparently suggest a violation of Onsager in 2 terminal setups, but I do not see currently any explanation of it. I think that inelastic dephasing effects and the asymmetric 2-terminal geometry may be the source of the breaking of Onsager, but this is speculative. As the Reviewer suggests, a 4-terminal measurement could contribute to shed light on this issue.

Response: We fully agree with the reviewer that the reconciliation of the Onsager relation with the ubiquitous observations of ‘linear-response’ magnetoconductance in two-terminal chiral spin valves remains an unresolved and vigorously debated issue in the CISS community. Importantly, while its theoretical interpretation continues to elicit controversy and await a definitive resolution, there is increasing evidence for the veracity of the experimental observations, as elaborated in our responses to Reviewer 3. Our proposal, while certainly not definitive, offers a self-consistent account of some of the most prominent (and most debated) observations in CISS spin valves (as summarized in the Conclusions section). Most importantly, we believe publication of reliable experimental results is most critical to the final resolution of this issue.

6): A minor correction:

In Fig. 2c I have the impression that the color code is not correct

"Pink squares are measured from I-V curves, whereas black circles are measured from MC measurements at different bias currents." For me the squares look grey and orange (and filled), the empty circles look blue-ish.

Response: We thank the reviewer for pointing this out. We have corrected the description of the symbols in the figure caption.

As a result, although the experimental part seems to me very well and accurately performed, I share the concerns of the Reviewer on the modelling part and on the rationalization of the experimental results. I can, therefore, not suggest the paper for publication in its present form.

Response: Again, we appreciate the positive assessment of our experimental work, especially the accuracy and veracity of the experimental results. We hope the responses, including those to Reviewer 3, have clarified the issues and alleviated the concerns regarding the modeling and rationalization of the experimental results.

Reviewers' Comments:

Reviewer #3:

Remarks to the Author:

I have read the rebuttal letter and the revised manuscript "Interplay of Structural Chirality, Electron Spin, and Topological Orbital in Chiral Molecular Spin Valves" by Y. Adhikari et al. My concern was that the previous version gave the impression that the experimental results violated the fundamental law of physics, the Onsager relation. In my opinion, the authors have addressed this issue satisfactorily in the revised version. The authors added a paragraph explaining a possible scenario reconciling the 'linear-response' magnetoresistance with the Onsager relation. The authors propose that the residual spin valve effect is due to a nonequilibrium effect caused by a charge trap. In Section 8 of the Supplementary Information, they provided experimental data showing hysteresis behavior in the I-V characteristic, which implies the existence of such a charge trap. Although, as the authors admitted, a definitive explanation is yet to be achieved, the paper would encourage the CISS community to investigate further an intriguing topic—the connection between the CISS effect and the Onsager relation, which holds fundamental importance in the quantum transport problem. In this regard, I believe the paper is worth considering for publication, possibly in Nature Communications.

Reviewer #4:

Remarks to the Author:

I think that the authors have addressed satisfactorily the issues I raised in my previous review. I am also glad to see that they have explained in a clearer way the issue of the magneto-chiral anisotropy. Although the issue of Onsager relations and time-reversal seems still not very clear, I believe that this cannot be fully addressed in the experimental study and will require a separate investigation. Therefore, I suggest the manuscript for publication.

Response to Reviewer #3:

I have read the rebuttal letter and the revised manuscript "Interplay of Structural Chirality, Electron Spin, and Topological Orbital in Chiral Molecular Spin Valves" by Y. Adhikari et al. My concern was that the previous version gave the impression that the experimental results violated the fundamental law of physics, the Onsager relation. In my opinion, the authors have addressed this issue satisfactorily in the revised version. The authors added a paragraph explaining a possible scenario reconciling the 'linear-response' magnetoresistance with the Onsager relation. The authors propose that the residual spin valve effect is due to a nonequilibrium effect caused by a charge trap. In Section 8 of the Supplementary Information, they provided experimental data showing hysteresis behavior in the I-V characteristic, which implies the existence of such a charge trap. Although, as the authors admitted, a definitive explanation is yet to be achieved, the paper would encourage the CISS community to investigate further an intriguing topic—the connection between the CISS effect and the Onsager relation, which holds fundamental importance in the quantum transport problem. In this regard, I believe the paper is worth considering for publication, possibly in Nature Communications.

Response: We are pleased that we were able to address the concern raised by the reviewer. We thank the reviewer for recommending our work to be published in Nature Communications.

Response to Reviewer #4:

I think that the authors have addressed satisfactorily the issues I raised in my previous review. I am also glad to see that they have explained in a clearer way the issue of magneto-chiral anisotropy. Although the issue of Onsager relations and time-reversal seems still not very clear, I believe that this cannot be fully addressed in the experimental study and will require a separate investigation. Therefore, I suggest the manuscript for publication.

Response: We are glad that the reviewer found the responses to the concerns satisfactory. We thank the reviewer for the constructive comments and support for publication.